# A self-regulated photothermal anti-/deicing film for all-season applications

Jiayu Du [1], Wenqi Wang [1], Yang Fu [1], Xin Li [1], Jie Tan [1], Hao Li[1], Xu Chen[1], Fuqiang Chu [2] ✉, Qi Min[3] & Chi Yan Tso [1] ✉

Ice accumulation poses a significant threat to aviation safety and energy infrastructure. Photothermal superhydrophobic surfaces offer a promising anti-/deicing strategy; however, their excessive heat absorption in summer accelerates material degradation and exacerbates urban heat island effects, highlighting the urgent need for dynamic thermal regulation. In this study, we present a self-regulated photothermal storage superhydrophobic film with a trilayer design, comprising a photothermal phase-change base layer, a freeze-resistant thermochromic hydrogel interlayer, and a transparent super-hydrophobic top layer. This multifunctional design enables seasonal adaptability, achieving 92% solar absorptance for efficient anti-/deicing in winter and 62% solar modulation to mitigate overheating in summer. This dual mode prolongs freezing time by 10-fold at −20 °C and lowers surface temperature by up to 17 °C in hot weather, demonstrating substantial potential for global building energy-savings. Additionally, its ultraviolet-blocking capability and durable superhydrophobicity ensure long-term durability performance in harsh environments. This work not only addresses the critical overheating challenge in photothermal materials but also advances the development of next-generation anti-icing systems.

Icing is a ubiquitous phenomenon that occurs in diverse forms, including frost, glaze, snow, and rime. While esthetically captivating, uncontrolled ice accretion presents severe hazards to critical infrastructure. In aviation, ice accretion compromises aerodynamic performance, tragically exemplified by the 2009 Colgan Air Flight 3407 crash that claimed 50 lives[1,2]. The 2021 Texas Winter Storm Uri demonstrated the vulnerability of power grids, where collapsed transmission lines resulted in 146 fatalities and US$195 billion in economic losses[3,4]. Additional impacts include up to 50% annual energy output reduction in wind turbines due to blade icing[5,6], and falling ice from buildings posing serious risks to pedestrians[7,8]. These incidents underscore the urgent need for efficient anti-/deicing technologies.

Traditional methods utilizing thermal, mechanical, or chemical energy face fundamental shortcomings like excessive energy consumption and environmental concerns[9–11]. In contrast, passive surface anti-icing technology (e.g., superhydrophobic surfaces[12], lubricant-infused surface[13], polyelectrolyte brushes[14]) offers promising alternatives by leveraging intrinsic surface properties without external energy input. Among various passive strategies, superhydrophobic surfaces have garnered widespread attention due to their remarkable water repellency, elevated ice nucleation barriers, and minimal ice adhesion[15,16]. However, these anti-icing surfaces can only delay ice formation but cannot inhibit it entirely, prompting researchers to explore hybrid technology combining passive anti-icing with active deicing. Photothermal superhydrophobic materials (PSMs) have emerged as a breakthrough strategy[17–19], converting solar energy into heat and elevating surface temperatures beyond ice melting points to enable gravity-assisted ice shedding. This combination maintains

[1]School of Energy and Environment, City University of Hong Kong, Tat Chee Avenue, Kowloon Tong, Hong Kong, China. [2]School of Energy and Environmental Engineering, University of Science and Technology Beijing, Beijing, China. [3]Key Laboratory of Advanced Reactor Engineering and Safety of Ministry of Education, Collaborative Innovation Center of Advanced Nuclear Energy Technology, Institute of Nuclear and New Energy Technology, Tsinghua University, Beijing, China. ✉e-mail: chufq@ustb.edu.cn; chiytso@cityu.edu.hk

surface dryness while preventing secondary icing. Apart from removing undesirable ice in aviation and energy applications, PSMs can reduce the heating energy consumption of buildings, demonstrating energy-saving performance in winter[20,21]. Nevertheless, their sunlight dependency limits effectiveness during cloudy conditions and at night. To address this constraint, photothermal storage superhydrophobic materials (PSSMs)[22–24] incorporating phase-change materials were developed to store daytime solar heat and release it at night for ice prevention.

However, extreme cold conditions do not exist all year round. The superior photothermal conversion capabilities of PSMs and PSSMs can even increase the surface temperature above 100 °C in summer[25,26], accelerating coating degradation and impairing sensitive components in energy and electrical systems. For example, wind turbines must operate below 80–120 °C to prevent thermal stress that shortens generator lifespan and rated capacity[27]. Moreover, the overheating penalty in summer intensifies thermal radiation, exacerbating urban heat island effects and elevating cooling energy consumption[28]. To overcome this critical flaw, adaptive materials with dynamic thermal regulation capability have been proposed, emerging as a promising solution. However, current research remains scarce and heavily relies on thermochromic microcapsule-based materials (TCMMs)[29,30], suffering from three key drawbacks: (i) insufficient solar absorptance (<90%) in cold conditions and narrow-band solar modulation (<50%, visible-light only), impairing photothermal deicing efficiency and thermal regulation capacity, (ii) inadequate hydrophobicity with contact angle below 120° and high phase change temperatures exceeding 20 °C that reduce anti-icing performance, and (iii) poor ultraviolet (UV) resistance that degrades leuco dyes and irreversibly diminishes thermochromism, thus hindering long-term outdoor applicability. Furthermore, several Janus-structure materials[31–35] with tunable radiative

cooling and solar heating have been proposed for season-adaptive thermal regulation, whereas they must be manually flipped and lack passive anti-icing capabilities.

To deal with these challenges, we propose a self-regulated photothermal storage superhydrophobic film featuring a rationally engineered trilayer structure (Fig. 1a). Diverging from conventional materials with constrained functionality, our design combines a broadband high-absorptance photothermal phase-change base layer with a freeze-resistant thermochromic hydrogel interlayer that exhibits suitable transition temperature and strong visible-near-infrared (VIS-NIR) modulation capability. A transparent superhydrophobic top layer is further integrated to prolong ice formation while maintaining optimal sunlight transmission for underlying photothermal conversion. This hierarchical strategy yields remarkable optical performance with 92% cold-state solar absorptance and 62% solar modulation ability, enabling efficient anti-/deicing in winter while passively mitigating overheating in summer. Such seasonal adaptation demonstrates global implications for building energy conservation, bridging a critical gap in year-round thermal management. More importantly, the UV-blocking and robust superhydrophobicity of the top layer ensures durability under harsh conditions, sustaining performance after 90-day outdoor exposure and 20 freeze-thaw cycles. By combining all the desired features within one single device, this work not only addresses the thermal management limitations of existing photothermal materials but also pioneers an energy-efficient and cost-effective next-generation anti/de-icing system for critical infrastructure and smart buildings.

## Results
### Design and fabrication of TAPSS film
Figure 1a illustrates the working principles of our temperature-adaptive photothermal storage superhydrophobic (TAPSS) film

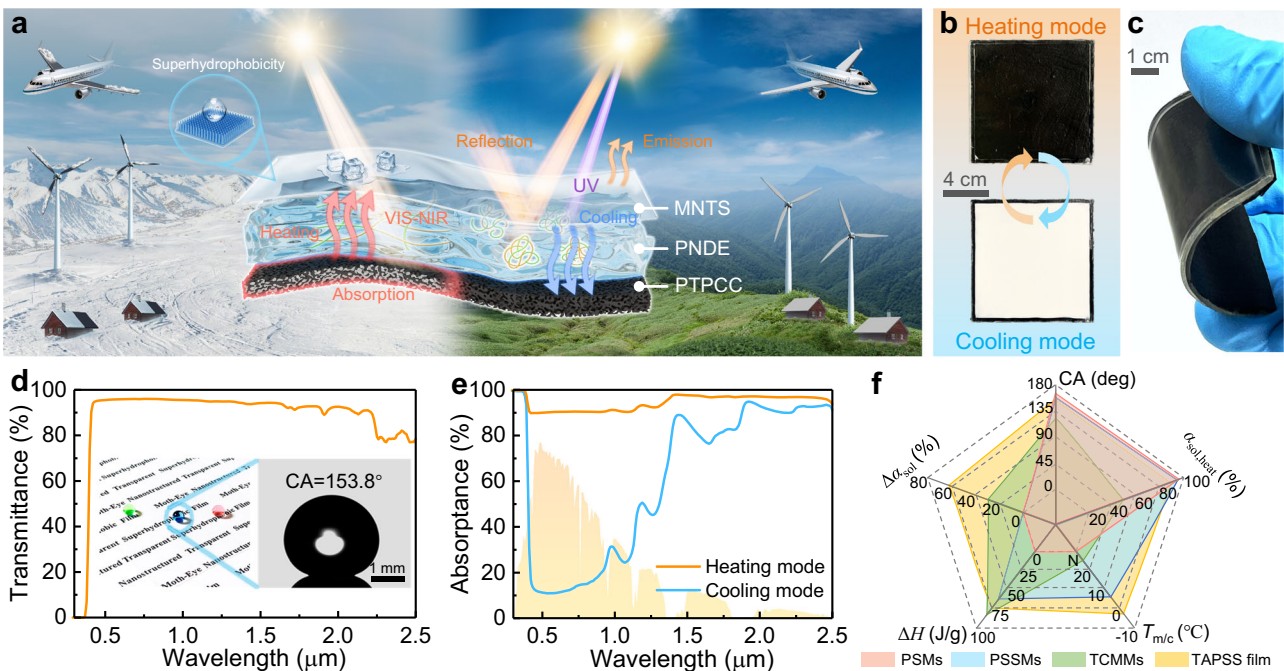

**Fig. 1 | Design and performance of the temperature-adaptive photothermal storage superhydrophobic (TAPSS) film. a** Schematic diagram depicting the trilayer structure working principle of TAPSS film. Highly-integrated configuration endows the TAPSS film with temperature-responsive dual modes: photothermal anti-/deicing in cold environments and radiative cooling in hot conditions. **b** Photographs of the TAPSS film (10 cm × 10 cm) in heating (20 °C) and cooling modes (40 °C). **c** Optical image demonstrating the flexibility under bending. **d** Transmittance spectrum and water contact angle of the MNTS film. **e** Absorption

spectra of the TAPSS film in heating and cooling modes. **f** Radar map comparing the performance of TAPSS film with photothermal superhydrophobic materials (PSMs), photothermal storage superhydrophobic materials (PSSMs), and thermochromic microcapsule-based materials (TCMMs) in five key metrics: contact angle (CA), solar absorptance modulation ($\Delta\alpha_{sol}$), heating-mode solar absorptance ($\alpha_{sol, heat}$), phase change enthalpy ($\Delta H$), and phase change temperature ($T_{m/c}$). The data originates from the averaged values from literature summarized in Supplementary Table 1. Source data are provided as a Source Data file.

consisting of a moth-eye nanostructured transparent super-hydrophobic (MNTS) top layer, a thermochromic poly(N-iso-propylacrylamide-co-N,N-dimethylacrylamide)/ethylene glycol (PNDE) hydrogel interlayer, and a photothermal phase change composite (PTPCC) base layer. This unique trilayer architecture integrates superhydrophobicity, thermochromism, photothermal conversion, and thermal energy storage, setting it apart from conventional pho-tothermal anti-/deicing materials. Particularly, the thermochromic property of PNDE hydrogel imparts a temperature-responsive dual mode to the TAPSS film, which appears black below the transition temperature to maximize solar absorption for photothermal anti-/deicing, while turning white above the transition threshold to reflect sunlight for radiative cooling (Fig. 1b). Coupled with remarkable flex-ibility (Fig. 1c), its intelligent optical adaptation allows deployment on complex curved surfaces.

To preserve the photothermal performance of the underlying layer while delaying ice formation, the top layer must simultaneously achieve high transparency and superhydrophobicity, which was fab-ricated via ultraviolet nanoimprint lithography (UV-NIL, Supplemen-tary Fig. 1) to precisely engineer subwavelength nanostructures. Using a chemically etched anodic aluminum oxide (AAO) template, moth-eye patterns[36] were replicated onto a UV-curable fluorinated acrylate substrate, yielding 90.9% solar transmittance ($\tau_{sol}$), 153.8° contact angle (CA), and 1.5° sliding angle (SA) (Fig. 1d, Supplementary Movie 1). Furthermore, the MNTS layer exhibits a sharp UV cutoff property with near-zero UV transmittance, effectively protecting underlying com-ponents from aging while maintaining high VIS-NIR transmittance to enhance photothermal conversion efficiency.

The adaptive functionality of TAPSS film necessitates a thermo-chromic interlayer that exhibits high transparency in the cold state, strong solar modulation capability, and a suitable transition tem-perature. Among various explored thermochromic materials, includ-ing vanadium dioxide (VO₂)[37], perovskite[38], supersaturated salt hydrate crystal[39], and liquid crystal elastomer[40], thermochromic hydrogels, particularly poly(N-isopropylacrylamide) (PNIPAM)-based systems[41], outperform due to strong solar modulation ability and thermal responsiveness. Here, we synthesized the PNDE hydrogel via one-pot in situ polymerization in an ice bath (Supplementary Fig. 2) using N-isopropylacrylamide (NIPAM) as the monomer, N,N-dimethylacry-lamide (DMAA) as the functional monomer, N-methylolacrylamide (MBA) as the crosslinker, ammonium persulfate (APS) as the initiator, and N,N,N',N'-tetramethylethylenediamine (TEMED) as the catalyst in a water/ethylene glycol (EG) binary solvent system. Notably, EG and DMAA synergistically tune the lower critical solution temperature (LCST) to the thermal comfort zone (22–28 °C)[42] while improving freeze resistance to preserve optical clarity in extreme cold conditions through hydrogen bond network modification. Such a suitable LCST is not only beneficial for mitigating urban heat island effects but also provides effective cooling protection for devices such as wind tur-bines, transmission lines, and photovoltaic panels. Moreover, the LCST can be readily tailored for different application scenarios by adjusting the ratios of DMAA and EG. Its thermochromism arises from temperature-dependent morphological changes. Below the LCST, the hydrogel maintains transparency through dominant hydrogen bond-ing between amide groups and water molecules that sustains polymer chain hydration. Above the LCST, hydrophobic isopropyl interactions induce phase separation, causing chain dehydration and aggregation into hydrophobic microdomains. This molecular rearrangement reduced the average pore diameter from 7.3 μm to 139 nm (Supple-mentary Fig. 3), significantly enhancing light scattering and driving the transparent-to-opaque transition.

The PTPCC base layer efficiently harnesses solar energy trans-mitted through the MNTS and PNDE layers, converting it into heat to facilitate ice melting. Simultaneously, the encapsulated PCMs store solar heat during daylight and release it under low-light conditions

(e.g., nighttime or overcast periods), thereby delaying freezing and compensating for the limitations of PSMs. This layer was fabricated via a two-step thermal curing process (Supplementary Fig. 4), comprising an upper multi-walled carbon nanotubes (MWCNTs)@poly-dimethylsiloxane (PDMS) photothermal (PT) layer that efficiently converts sunlight into thermal energy while maintaining optimal sur-face darkness during PCM solidification to prevent photothermal performance degradation (Supplementary Fig. 5), and a lower PCM/MWCNTs@PDMS composite phase change material (CPCM) layer for thermal energy storage. In this configuration, the MWCNTs enable broadband solar absorption and high photothermal conversion effi-ciency through thermal molecular vibration, while the PDMS matrix encapsulates the PCM and ensures structural flexibility (Supplemen-tary Fig. 6). To maximize anti-icing performance, we employed a binary PCM system with tailored phase change temperatures and enthalpies.

To ensure robust interfacial bonding among the multilayer com-ponents, several strategies were employed. The polyethylene ter-ephthalate (PET) substrate, which serves both as the encapsulation layer for the hydrogels and the bonding interface with the PTPCC layer, was plasma-treated to promote adhesion. The curing agent ratio of PDMS was further optimized to strengthen its bonding with the PET substrate. Additionally, adhesive tape and UV-curable glue were applied along the edges to reinforce adhesion and prevent hydrogel leakage. Peel tests confirmed that the MNTS film and PNDE hydrogel were effectively bonded with a maximum peel strength of 384 N/m, while the adhesion between the PET film and PTPCC layer reached 78 N/m (Supplementary Fig. 7). The trilayer synergy empowers the TAPSS film with a high solar absorptance of 92% in heating mode and a low value of 30% in cooling mode, yielding a remarkable solar mod-ulation of 62% (Fig. 1e). Comparative analysis with conventional pho-tothermal anti-/deicing materials reveals its outstanding performance across five key aspects including contact angle (CA) governing anti-icing capacity, solar absorptance in heating mode ($\alpha_{sol, heat}$) dominat-ing deicing efficiency, solar modulation ability ($\Delta\alpha_{sol}$) reflecting the difference in heating and cooling performance, phase change enthalpy ($\Delta H$) and phase transition temperature ($T_{m/c}$), both determining the thermal energy storage capacity (Fig. 1f, Supplementary Table 1). Notably, beyond having comparably high CA and $\alpha_{sol, heat}$ to PSMs and PSSMs, our TAPSS film demonstrates unprecedented $\Delta\alpha_{sol}$ among all photothermal anti-/deicing strategies, highlighting its superior ther-mal management capability. Additionally, it combines optimal thermal energy storage capacity ($\Delta H$-75 J/g, $T_{m/c}$--2.4 °C) with robust super-hydrophobicity, conferring prolonged anti-icing performance. More importantly, the UV shielding top layer delivers exceptional durability to the TAPSS film without significant degradation in optical perfor-mance and superhydrophobicity after continuous 90-day outdoor tests (Supplementary Fig. 8).

## Superhydrophobicity, antireflection, and stability of MNTS film
The MNTS film exhibited superhydrophobicity as water droplets rebounded rapidly upon impact with a contact time ($t_c$) of 13.3 ms (Fig. 2a, Supplementary Movie 2), aligning well with the theoretical Rayleigh limit[43,44]

$$t_c = 2.22\sqrt{\rho R_0^3/\gamma} = 12.9 \text{ ms} \tag{1}$$

where $\rho$, $R_0$, and $\gamma$ represent the density, radius, and surface tension of the droplet, respectively. This superhydrophobicity enabled effective self-cleaning, where incoming droplets readily rolled off while removing surface particles (Supplementary Movie 3), thereby main-taining high transparency during long-term outdoor operation. The water-repellent properties originate from synergistic effects of precisely engineered nanostructures and chemical composition. On the one hand, the moth-eye nanostructures (Fig. 2b) enhance hydrophobicity by stabilizing air pockets beneath droplets,

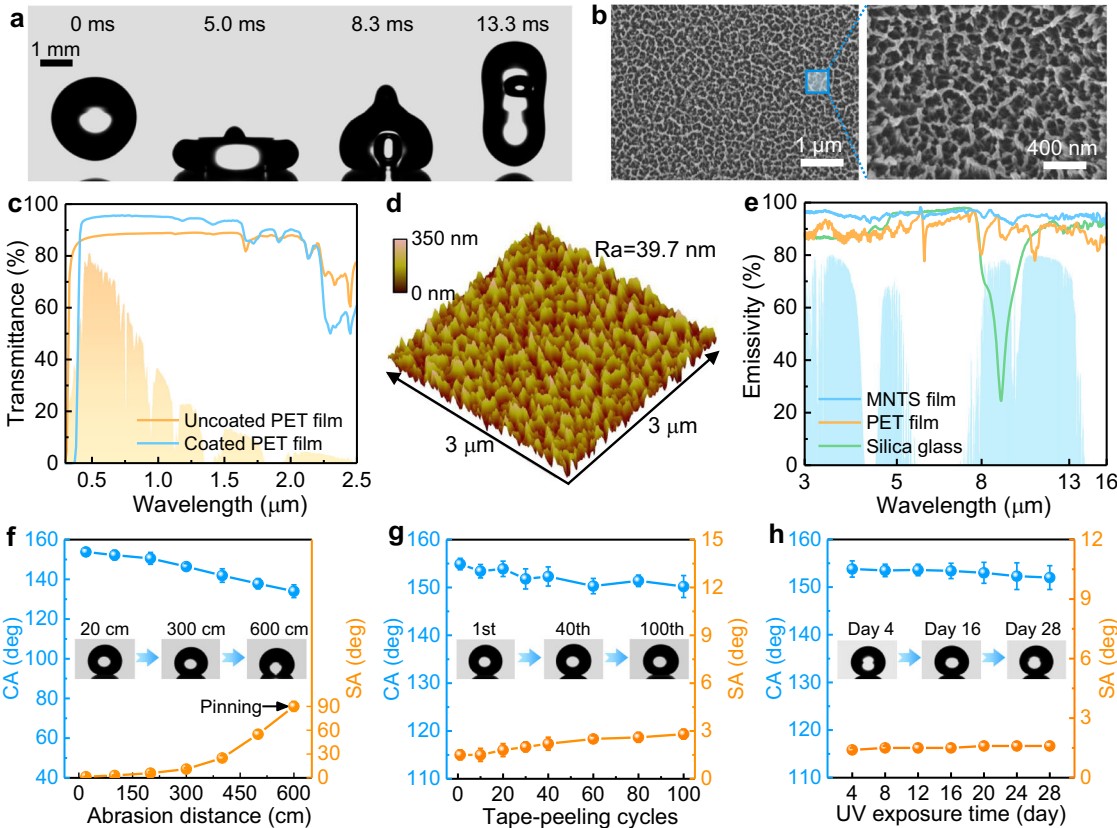

**Fig. 2 | Microscopic morphology, antireflection and superhydrophobic stability of the moth-eye transparent superhydrophobic (MNTS) film. a** Snapshots of a 10 μL water droplet impingement at an impact velocity of 0.5 m/s. **b** Scanning electron microscopy (SEM) images of the MNTS film. The SEM measurements were independently repeated three times with consistent results. **c** Transmittance spectra of MNTS-coated and uncoated PET films. **d** Atomic force microscopy (AFM) image. **e** Infrared emissivity spectra of polyethylene terephthalate (PET) film, silica glass, and MNTS film. Changes in contact angle (CA) and sliding angle (SA) during **f** sandpaper abrasion, **g** tape-peeling, and **h** UV aging tests. Dot heights, mean values; error bars, standard deviation ($n = 5$). Source data are provided as a Source Data file.

establishing a robust low-adhesion Cassie-Baxter state[45]. On the other hand, XPS analysis (Supplementary Fig. 9) revealed high fluorine content (41.6 at%), confirming abundant fluorinated groups in the UV-curable acrylate that minimized surface energy. High resolution spectra identified $CF_3$ (294.5 eV), $CF_2$ (293.0 eV), C=O (288.4 eV), C-O (286.0 eV) and C-C/C-H (284.8 eV) in the C $1s$ spectrum[46–48] along with $CF_3$ (689.4 eV) and $CF_2$ (688.5 eV) in the F $1s$ spectrum[19,49] In addition to particles, the chemical contamination resistance was further evaluated using sodium dodecyl sulfate (SDS) and tetradecane. The surface maintained a high CA (>150°) and low SA (<10°) when the SDS concentration was below 0.5 critical micelle concentration (CMC) (Supplementary Fig. 10). In contrast, the MNTS film exhibited only moderate oleophobicity toward tetradecane (CA ~ 95°), and therefore cannot provide self-cleaning against oil contamination. Nevertheless, after rinsing, the surface retained stable superhydrophobicity after being immersed in tetradecane for 120 h.

The MNTS film achieved dual optical advantages, combining excellent transparency with antireflection properties. MNTS-coated PET showed much higher transmittance ($\tau > 95\%$) than bare PET ($\tau \sim 88\%$, Fig. 2c) due to biomimetic moth-eye nanostructures[50,51] that lowered reflectivity from 11.8% to 4.3% (Supplementary Fig. 11). This antireflection is attributed to three key characteristics, including nanoscale surface roughness (~39.7 nm, Fig. 2d), periodically arranged subwavelength nanostructures (~100 nm height, ~150 nm spacing; Supplementary Fig. 12), and an intermediate refractive index ($n$~1.36, Supplementary Fig. 13). Together, these features created a smooth refractive index gradient from air ($n \sim 1.0$) to PET ($n \sim 1.55$), effectively suppressing Fresnel reflections[52] Furthermore, the MNTS film

demonstrated a broadband high emissivity of 94.5% within the atmospheric window (8–13 μm, $\varepsilon_{8-13}$) (Fig. 2e), contributing to radiative cooling in hot weather, whereas the $\varepsilon_{8-13}$ of conventional molds for encapsulating hydrogels (e.g., silica glass[53] and PET[54]) was 76.4% and 89.1%, respectively. This high value arose from the collective and cumulative effects of C–O stretching vibrations, C-H bending vibrations, C–F stretching and bending vibrations in the range of 770–1250 $cm^{-1}$[55,56], as evidenced by the attenuated total reflectance-Fourier transform infrared spectroscopy (ATR-FTIR) spectrum in Supplementary Fig. 14.

The MNTS film serves as a protective layer for the underlying PNDE hydrogel and PTPCC layers, necessitating robust mechanical durability, UV resistance, and chemical stability. To evaluate its mechanical durability, sandpaper abrasion and tape-peeling tests were conducted. As shown in Fig. 2f and g, the superhydrophobicity (CA > 150° and SA < 10°) was retained after 200 cm sandpaper abrasion and 100 tape-peeling cycles. This performance significantly surpassed commercial Glaco coatings, which failed after just 30 cm abrasion due to poor $SiO_2$ nanoparticle-substrate adhesion (Supplementary Fig. 15). Our film also withstood continuous manual abrasion (Supplementary Movie 4), high-pressure water jets (Supplementary Movie 5), sand impact of 1000 g and acid rain exposure for 2 hours (Supplementary Fig. 16), while the superhydrophobicity of Glaco coatings was easily compromised under gentle handling. These tests collectively demonstrate the robust mechanical durability of MNTS film. Accelerated UV aging test showed that the MNTS film exhibited excellent UV resistance, retaining superhydrophobicity after 4 weeks of intense UV exposure (0.89 W/m² at 340 nm; equivalent to 8 months of Florida

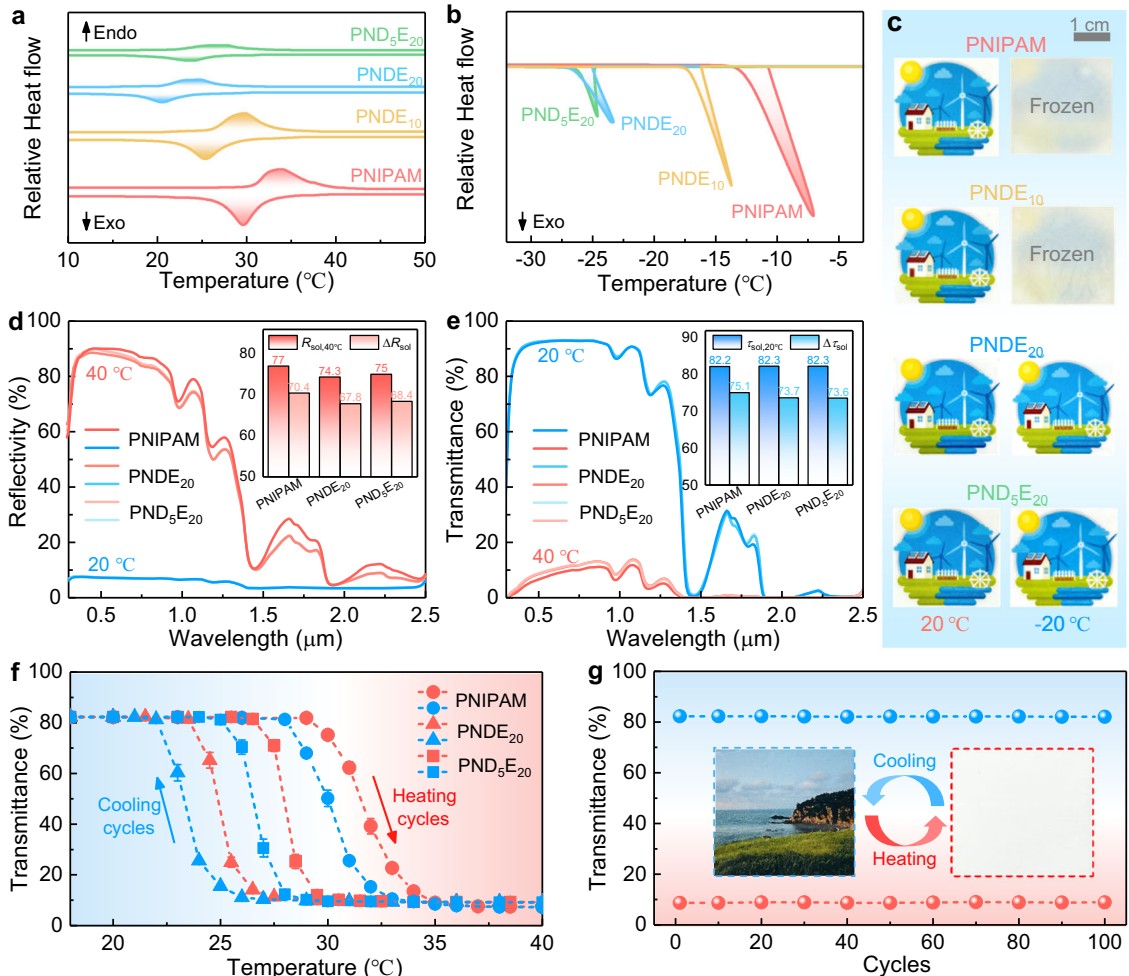

**Fig. 3 | Optical performance and freeze resistance of thermochromic poly(N-isopropylacrylamide-co-N,N-dimethylacrylamide)/ethylene (PNDE) glycol hydrogels.** Differential scanning calorimeter (DSC) diagrams for **a** lower critical solution temperature (LCST) and **b** freezing point. **c** Photographs of hydrogel-covered picture in the 20 °C (left) and −20 °C (right) environment. **d** Solar reflectivity at 40 °C and **e** solar transmittance spectra at 20 °C. Insets show the values of solar reflectivity at 40 °C ($R_{sol,\,40\,°C}$), reflectivity modulation ($\Delta R_{sol}$), solar transmittance at 20 °C ($\tau_{sol,20\,°C}$), and transmittance modulation ($\Delta \tau_{sol}$). **f** Solar transmittance as a function of temperature. Dot heights, mean values; error bars, standard deviation ($n = 3$). **g** Solar transmittance of $PND_5E_{20}$ hydrogels measured after 100 heating-cooling cycles. Insets show the photographs of the $PND_5E_{20}$ hydrogel-covered picture in the cold state (left) and hot state (right). Source data are provided as a Source Data file.

sunlight[57], Fig. 2h). This durability stems from robust C-F bonds (bond energy: 485 kJ/mol) that resist UV degradation[58]. Chemical stability tests further confirmed its robustness, with maintained performance after 24-hour immersion in acidic (pH=2), alkaline (pH=12), and saline (1 M NaCl) solutions (Supplementary Fig. 17). Notably, the MNTS film combines these durable superhydrophobic characteristics with excellent VIS-NIR transmittance and UV resistance compared to existing transparent superhydrophobic surfaces (Supplementary Table 2). These unique properties ensure reliable long-term outdoor operation of the integrated TAPSS film.

**Optical performance and freeze resistance of PNDE hydrogel**
The PNDE hydrogel layer serves as the key component of the TAPSS film for solar modulation. Through comprehensive optimization, we determined that a 2 mm-thick hydrogel containing 20 wt% NIPAM enables an optimal balance between optical transmission and modulation ability, achieving 82.2% solar transmittance ($\tau_{sol,20\,°C}$) in the transparent state while maintaining 77.0% solar reflectivity ($R_{sol,40\,°C}$) in the opaque state (Supplementary Fig. 18). Here, all optical measurements were conducted using silica glass encapsulation to eliminate potential interference from UV-absorbing MNTS films.

The limited applicability of native PNIPAM hydrogel, stemming from its elevated LCST (29.6–33.9 °C, Fig. 3a) beyond the thermal comfort range (22–28 °C) and inadequate freeze resistance (−7.1 °C) (Fig. 3b, c), was addressed through the development of a water/EG binary solvent system. EG incorporation modulates the LCST by reducing solvent polarity and weakening hydrogel-solvent hydrogen bonding. The formation of stronger EG-water hydrogen bonds decreases free water content, simultaneously depressing the freezing point. Compared to other alcohol-based solvents like ethanol, glycerol, and PEG400[59–62], EG delivered three critical improvements under 15 wt% condition: enhanced freeze resistance (−18 °C), suitable LCST (23.1–27.5 °C), and excellent hot-state reflectivity ($R_{sol,40°C}$ = 75.3%) (Supplementary Fig. 19 and 20). As the temperatures in winter frequently plunge below −20 °C in high-latitude regions, we further optimized the EG content to enhance freeze resistance. Systematic evaluation revealed that increasing EG concentration progressively lowered both the LCST and freezing point (Supplementary Fig. 21). The $PNDE_{20}$ formulation (20 wt % EG) achieved a remarkable freezing point (−23.3 °C), maintaining high transparency at −20 °C (Fig. 3b, c). Although this modification caused minor reductions in $R_{sol,\,40\,°C}$, and reflectivity modulation ($\Delta R_{sol}$), it preserved satisfying $\tau_{sol,\,20\,°C}$ (>82%, Fig. 3d, e).

However, this EG concentration depressed the LCST to 20.6–24.8 °C below the thermal comfort zone (Fig. 3a, Supplementary Fig. 21), necessitating further optimization. Accordingly, we developed P(NIPAM-co-DMAA) hydrogels by incorporating the hydrophilic monomer DMAA. The optimized $PND_5E_{20}$ hydrogel achieved three critical improvements: (i) restoration of the ideal LCST (24.0–27.8 °C) through enhanced hydrophilicity (Fig. 3a, Supplementary Fig. 22), (ii) strong freeze resistance (−24.6 °C freezing point) via strengthened polymer-water hydrogen bonding (Fig. 3b), and (iii) superior thermo-responsive behavior with sharp optical transitions (<3 °C window), minimal hysteresis (1.5 °C between heating-cooling cycles, Fig. 3f) and rapid switching kinetics (9.7 s for heating and 7.5 s for cooling; Supplementary Fig. 23 and Movie 6). It is observed that the transition temperatures estimated through Fig. 3f showed good agreement with DSC measurements.

Notably, DMAA modification preserved optical performance, as the $PND_5E_{20}$ hydrogel exhibited $\tau_{sol,20\,°C} = 82.3\%$ ($\tau = 92.5\%$ in the visible range) and $R_{sol,40\,°C} = 75.0\%$ ($R = 88.1\%$ in the visible range), corresponding to a solar modulation ability comparable to the $PNDE_{20}$ hydrogel (Fig. 3d, e). The hydrogel demonstrated exceptional durability without obvious degradation in optical performance after 100 thermal cycles (Fig. 3g), attributable to effective encapsulation that limited water loss to <1% versus 64.6% in unsealed controls after heating for 10 hours (Supplementary Fig. 24). As benchmarked in Supplementary Table 3, the $PND_5E_{20}$ hydrogel combines multiple superior properties including excellent solar modulation ability ($\Delta\tau_{sol} = 73.6\%$, $\Delta R_{sol} = 68.4\%$), extreme freeze resistance (−24.6 °C), precisely tuned LCST ( ~ 26 °C), and ultra-fast response (~0.15 min). Its unprecedented reflectivity modulation, often overlooked in previous studies, surpasses reported thermochromic hydrogels (Supplementary Fig. 25), resulting from the optimized thickness and NIPAM concentration. These capabilities jointly establish $PND_5E_{20}$ hydrogel as the ideal interlayer for our TAPSS film, providing seasonal thermal management through efficient solar heating in winter and heat blocking in summer.

## Photothermal performance and thermal energy storage capacity of PTPCC

The photothermal performance of the PTPCC base layer determines the deicing efficiency of the TAPSS film, which was optimized by tuning the MWCNTs loading. Even at 2 wt% MWCNTs content, a broadband solar absorptance ($\alpha_{sol}$) of nearly 97% was achieved (Fig. 4a) due to continuous energy levels of hybrid bonds[63]. Under 1.0 sun illumination (1 kW/m²), the PT film demonstrated rapid heating from 23 °C to over 60 °C within 200 s, stabilizing at ~65 °C with ~90% photothermal conversion efficiency (Fig. 4b, c and Supplementary Note 1), mediated by π-π* transition-induced thermal vibrations[11]. This performance significantly surpassed PDMS with $\alpha_{sol} = 3.58\%$ and a 30 °C equilibrium temperature (Supplementary Fig. 26). While increased MWCNT loading (up to 6 wt%) increased heating rates and photothermal conversion efficiency due to improved thermal conductivity (Fig. 4c), further enhancement was marginal, potentially compromising thermal storage capacity and mechanical strength. Thus, PT-6 was chosen as the optimal formulation, achieving an equilibrium temperature of 65.1 °C and a photothermal conversion efficiency of 91.2%. Infrared thermography confirmed PT-6's photothermal performance (Fig. 4d), with surface temperature exceeding bottom temperature measured by thermocouples. In addition, it exhibited excellent stability with maintained photothermal performance after six heating-cooling cycles (Fig. 4e).

Enhancing the thermal storage capacity of PTPCC layer is crucial for TAPSS film, enabling synergistic cooperation with the MNTS top layer to further delay icing, especially in sunlight-limited conditions. Given that water droplets on superhydrophobic surfaces typically resist freezing above −10 °C due to nucleation suppression, we

targeted PCMs with phase transitions between −10° and 0 °C for effective icing delay[24]. Unlike conventional anti-icing PCMs, which often operate at 0–10 °C[22,64,65] and suffer from premature latent heat release, we identified n-tridecane as an ideal candidate due to its phase transition near −6 °C (Supplementary Figs. 27 and 28). However, to compensate for its moderate enthalpy (-155 J/g), we engineered a binary alkane system by blending n-tridecane with higher-capacity n-tetradecane (-208 J/g). Through systematic optimization, a 1:3 n-tridecane/n-tetradecane ratio presented the best performance, achieving precisely tailored thermal properties: melting at 1.08 °C ($\Delta H_m = 195.08$ J/g) and crystallizing at −3.85 °C ($\Delta H_c = 195.45$ J/g). This optimized binary PCM not only minimizes supercooling under practical freezing conditions but also maximizes thermal storage capacity.

Through systematic optimization of PCM content in CPCMs, we established an optimal compromise between thermal storage capacity and leakage resistance. Increasing PCM loading from 60 to 120 wt% (relative to PDMS) enhanced phase change enthalpy from 61 to 93 J/g while maintaining stable phase change temperatures (Fig. 4f, g, Supplementary Fig. 29). However, this improvement was accompanied by increased PCM leakage (rising from 5.1 to 9.3 wt%, Supplementary Fig. 30), with visible substrate contamination occurring above 80 wt% loading. The leakage resistance results from the uniform dispersion of MWCNTs within the PDMS matrix, which creates a nanoporous network that effectively immobilizes PCM through strong capillary forces (Fig. 4h). Consequently, the CPCM-80 formulation (80 wt% PCM) emerged as the optimal composition, exhibiting well-balanced phase transition temperatures ($T_m = 1.50$ °C, $T_c = -6.34$ °C), excellent phase change enthalpy ($\Delta H_m = 75.01$ J/g, $\Delta H_c = 75.09$ J/g) corresponding to encapsulation performance (encapsulation efficiency $E_{en} = 38.45\%$, latent heat storage efficiency $E_{es} = 38.43\%$, and thermal storage capacity $C_{es} = 99.96\%$ calculated via Eqs. S4-6, Supplementary Table 4).

CPCM-80 maintained excellent phase change reversibility and thermal stability through 100 melting-solidification cycles, with negligible variations in transition temperatures and enthalpies (Supplementary Fig. 31). Consistent with classical nucleation theory (Supplementary Note S2), the observed supercooling ($\Delta T = T_m - T_c$) in CPCMs exceeded that of pure PCM, resulting from the enhanced hydrophobicity that increases the energy barrier of heterogeneous nucleation (Fig. 4g, Supplementary Fig. 32). To evaluate the delayed cooling performance, we conducted comparative temperature analyses of PT-6, CPCM-80, and PTPCC under alternating light conditions in a cold environment (Fig. 4i), where the superhydrophobic and thermochromic layers were not applied. Our results demonstrate that both CPCM-80 and PTPCC effectively prolonged the cooling process, delaying the temperature drop to −10 °C by 540 s and 640 s, respectively, compared to PT-6. This enhanced thermal buffering stems from latent heat release during phase transition, enabling effective anti-icing performance without solar energy input.

## Anti-/deicing and defrosting performance

The anti-icing test was conducted in a cold chamber (−20 °C, 20% relative humidity (RH)), where the freezing processes of 10 μL water droplets were recorded by a high-speed camera (Supplementary Fig. 33). The results show that water droplets froze within 108 s on PET and 182 s on PTPCC, while the TAPSS film demonstrated exceptional freezing delay (1115 s) (Fig. 5a, Supplementary Movies 7 and 8), representing a 10-fold improvement attributed to the surface superhydrophobicity reducing heat transfer area and elevating heterogeneous nucleation energy barriers (Supplementary Fig. 34 and Note S2). The icing process involved two sequential stages: nucleation-recalescence featuring dendritic ice crystal formation and microbubble generation that rendered droplets opaque (Fig. 5a, b); and final complete freezing with upward-propagating ice fronts forming trapped bubbles, creating characteristic pointed tips through ice expansion and surface tension effects[66,67]. Remarkably, TAPSS film exhibited

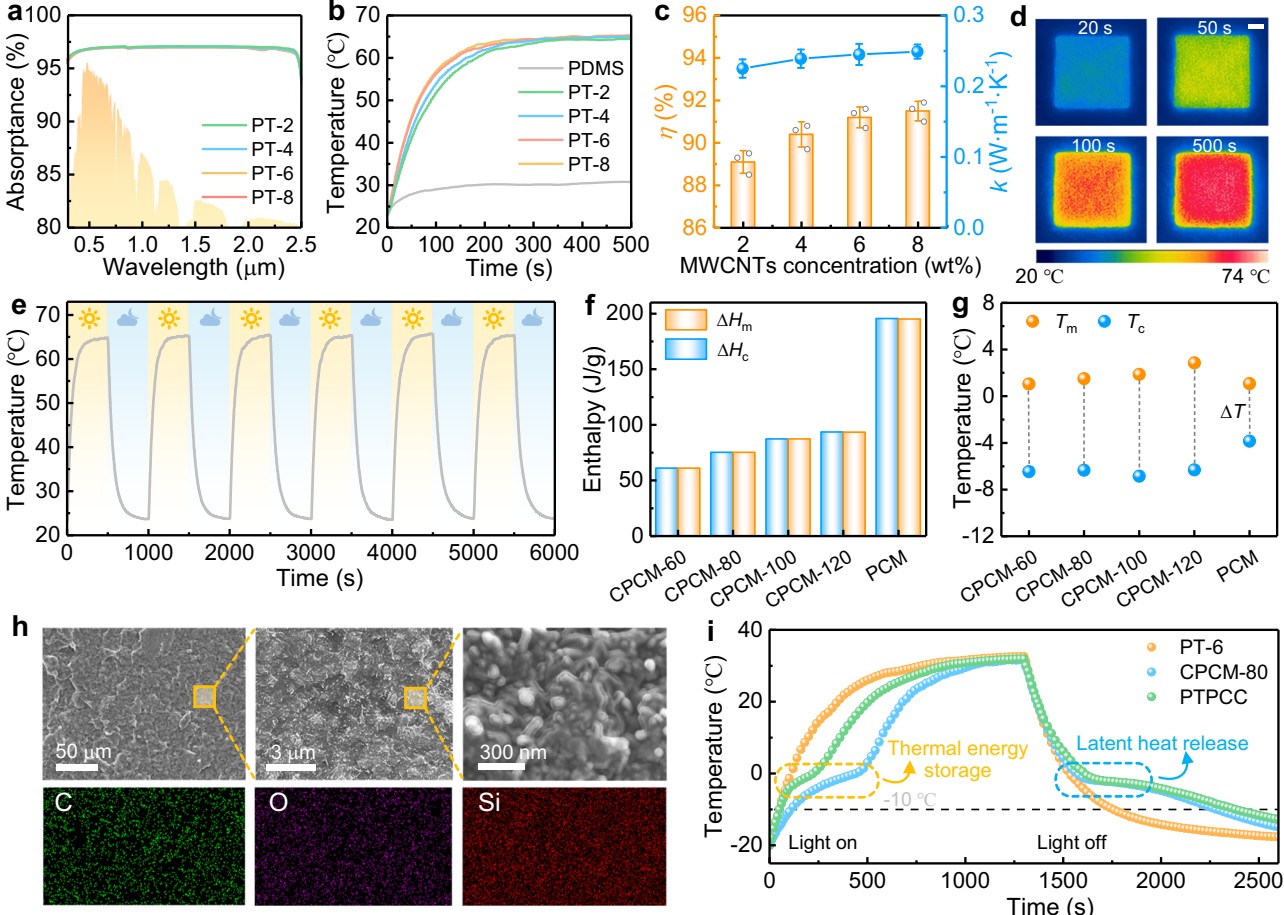

**Fig. 4 | Light absorption, photothermal performance and thermal energy storage capacity of photothermal phase change composite (PTPCC). a** Solar absorptance spectra, **b** Temperature evolution curves, **c** photothermal conversion efficiency ($\eta$) and thermal conductivity ($k$) for photothermal (PT) films with different multi-walled carbon nanotubes (MWCNTs) concentration. Dot and bar heights, mean values; error bars, standard deviation ($n = 5$). **d** Infrared images of PT-6 surface temperature under 1.0 sun illumination. The scale bar is 1 cm. **e** Temperature evolution curves of PT-6 during six heating-cooling cycles. **f** Phase change temperature and **g** phase change enthalpy for composite phase change materials (CPCMs) with different mass ratio of phase change material (PCM) to polydimethylsiloxane (PDMS). **h** Scanning electron microscopy (SEM) images and energy dispersive spectrometer (EDS) mapping of the cross section of photothermal phase change composite (PTPCC). The SEM and EDS measurements were independently repeated three times with consistent results. **i** Temperature evolution curves of PT-6, CPCM-80 and PTPCC with the light on and off (−20 °C, 20% RH, 1.0 sun illumination), where the plateaus indicate the thermal energy storage and the latent heat release. Source data are provided as a Source Data file.

outstanding dynamic anti-icing performance, as evidenced by continuous bouncing and sliding of water droplets on 3° inclined surfaces (Supplementary Movie 9).

The photothermal deicing performance of TAPSS film was evaluated under 1.0 sun illumination (−20 °C, 20% RH). Figure 5c and Supplementary Movie 10 show ice whiskers gradually melting as the temperature increased, with complete ice droplet melting occurring at ~410 s, in stark contrast to the MNTS film, which showed negligible melting after 1000 s due to its minimal photothermal effect. The melting process initiated at 0 °C with a bottom-up progression, revealing a distinct ice-water front by 320 s. During melting, Marangoni-effect-driven bubble migration from non-melting to melting zones promoted Cassie state recovery[67,68] (Fig. 5b). Compared to conventional photothermal anti-/deicing materials (Supplementary Fig. 35 and Table 5), TAPSS film demonstrated superior icing delay but slower melting, providing an effective thermal buffer against temperature rise and heat stress. This phenomenon can be explained by three mechanisms: (i) reduced solar energy transmitting to the PTPCC layer due to the existence of MNTS film and PNDE hydrogel, (ii) thermal energy storage by PCM rather than heat conduction to the ice droplet, and (iii) increased thermal resistance from the non-contact

configuration between the ice droplet and the PTPCC layer (Supplementary Fig. 36 and Note S3). The thermal conductivity of hydrogel can be further improved by introducing nanoparticles with high thermal conductivity and constructing highly ordered hierarchical structures. The deicing performance was further evaluated using ice cubes placed on the roof surface of a house model. On the TAPSS-coated side, the ice cube melted and slid downward under gravity at 555 s, whereas on the uncoated side, the ice cube remained unmelted and firmly adhered to the surface (Fig. 5d). Notably, the TAPSS film retained its excellent superhydrophobicity and optical properties after 20 freeze-thaw cycles (Supplementary Fig. 37). Although these results demonstrate promising initial durability, further improvement is necessary to meet the long-term service requirements for practical applications, which may span several years.

Frost accumulation under low-temperature, high-humidity conditions presents critical challenges for aircraft and wind turbine operations, where it degrades surface superhydrophobicity and alters aerodynamic profiles. To quantify the defrosting performance of our TAPSS film, we conducted controlled experiments in an environmental chamber (−15 °C, 80% RH) using: (1) an aircraft model with asymmetrically treated wings (TAPSS-coated vs. uncoated), and (2) a wind

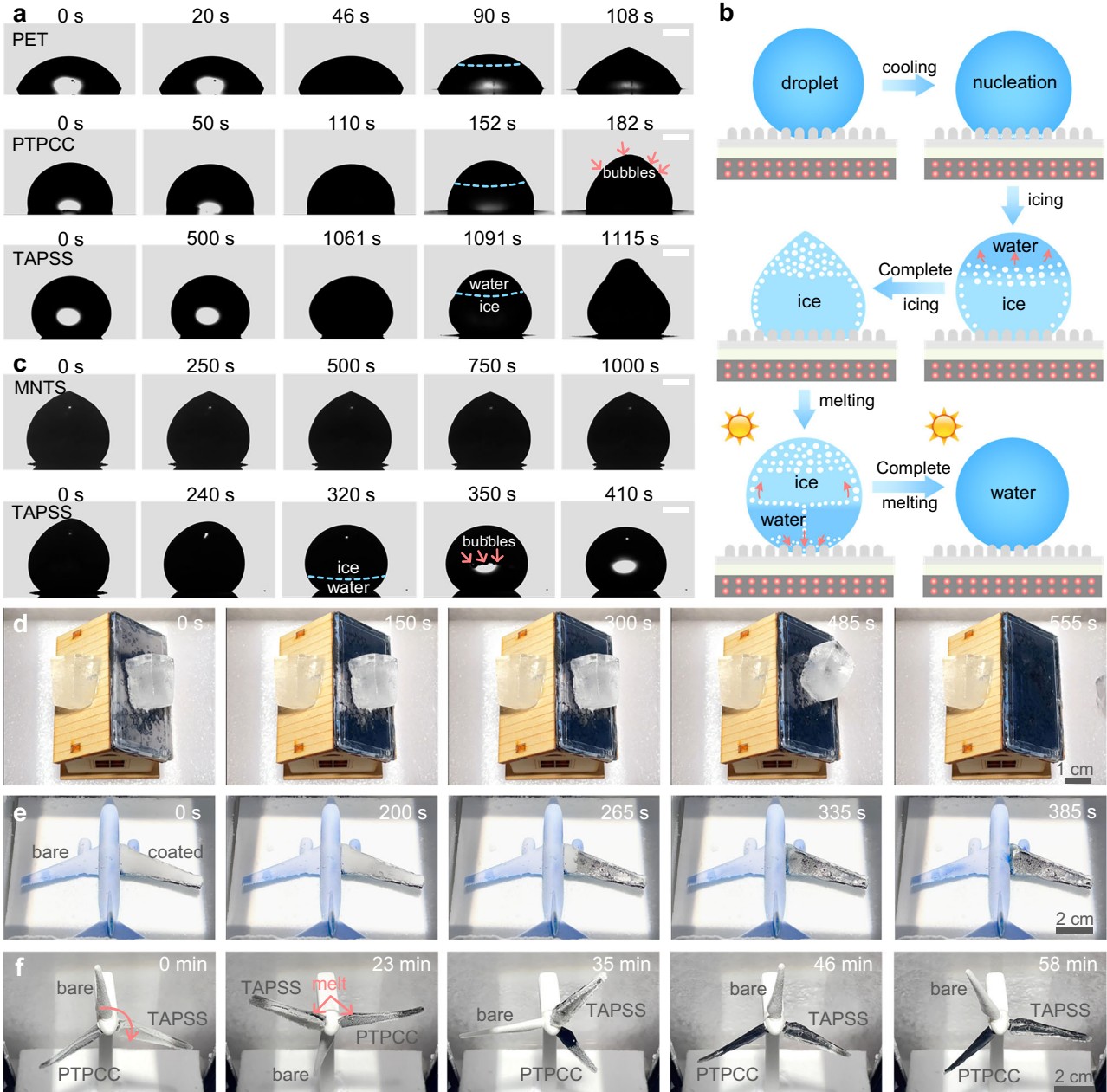

**Fig. 5 | Anti-icing, deicing and defrosting performance of temperature-adaptive photothermal storage superhydrophobic (TAPSS) film. a** Freezing process of 10 µL water droplets on polyethylene terephthalate (PET), photothermal phase change composite (PTPCC) and TAPSS films at −20 °C and 20% RH. The scale bars are 1 mm. **b** Schematic diagram of icing and melting processes of droplets on TAPSS film. **c** Melting process of 10 µL ice droplets on moth-eye transparent superhydrophobic (MNTS) and TAPSS films (−20 °C, 20% RH). The scale bars are 1 mm. **d** Deicing process of ice cubes (1 cm×1 cm × 2 cm) on the roof surfaces of a model house with one side coated with TAPSS film (−20 °C, 20% RH). Defrosting processes on the **e** aircraft model and **f** rotating wind turbine model (−15 °C, 80% RH). The simulated solar intensity is 1.0 sun illumination for all cases. Source data are provided as a Source Data file.

turbine model with three distinct blade treatments (TAPSS-coated, PTPCC-coated, and uncoated control). Under 1.0 sun illumination, we observed that the TAPSS-coated aircraft wing initiated frost melting at 200 s and achieved complete clearance by 385 s, while uncoated surfaces showed negligible melting (Fig. 5e, Supplementary Movie 11). The rotating wind turbine blades (16 rpm) exhibited more complex defrosting dynamics (Fig. 5f, Supplementary Movie 12), with melting initiating at 23 minutes from the leading/trailing edges before propagating rootward, while tips remained frosted until 58 minutes. This significantly delayed response stems from three key factors: (i) vertical blade orientation limiting solar absorption to edges, (ii) rotational motion preventing sustained solar exposure, and (iii) rotation-enhanced convective cooling. Notably, while PTPCC-coated blades showed earlier root melting (visible at 35 min), their complete defrosting time matched TAPSS-coated blades, demonstrating that edge-focused photothermal effects, rather than uniform surface heating, govern the defrosting process in rotating systems. Durability tests of the TAPSS film under acid rain and sand impact (Supplementary Fig. 16) further confirmed its potential for dynamic applications such as wind turbine blades and aircraft wings.

## Cooling performance and energy-saving evaluation
Unlike conventional photothermal materials, our TAPSS film not only enables efficient anti-/deicing in winter but also prevents overheating

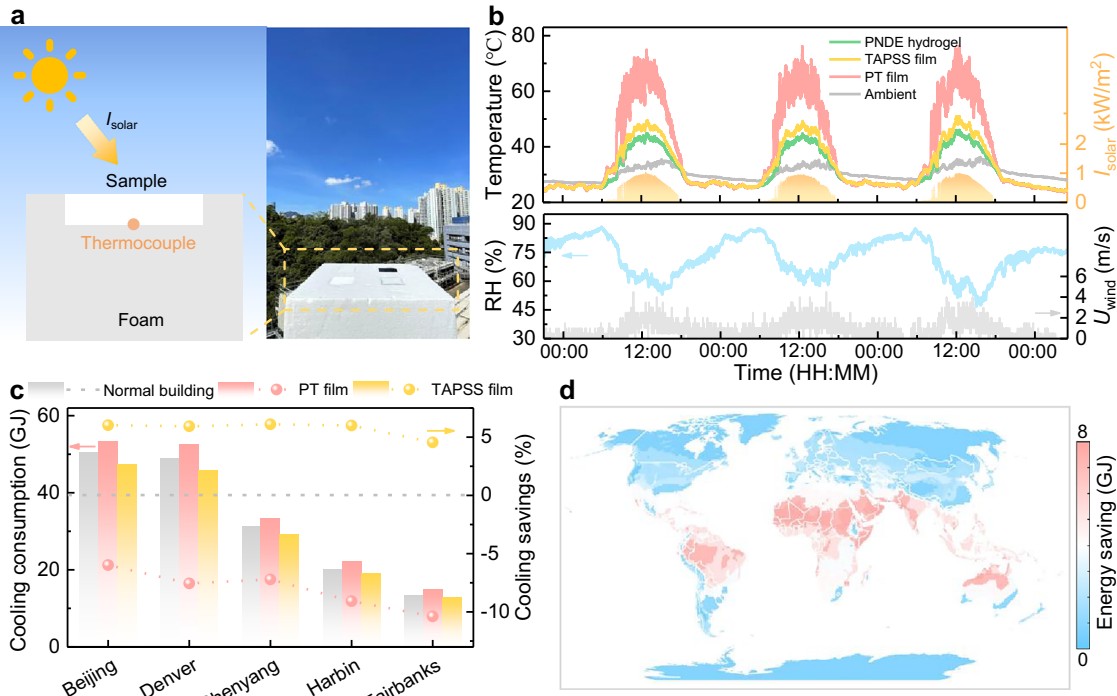

**Fig. 6 | Outdoor cooling performance and global energy-saving evaluation of temperature-adaptive photothermal storage superhydrophobic (TAPSS) film. a** Schematic of the outdoor field test setup. **b** Continuous 80-hour (6–10 June 2025) field measurements in Hong Kong showing temporal variations of sample temperature, ambient air temperature, solar intensity ($I_{solar}$), wind speed ($U_{wind}$), and relative humidity. **c** Annual cooling energy consumption and savings for buildings equipped with TAPSS and photothermal (PT) roofs, and normal buildings across cities in various climate zones. **d** Global energy-saving potential maps showing net energy consumption differences between TAPSS-equipped and conventional buildings. Source data are provided as a Source Data file.

in summer due to its strong solar regulation capability. This dual functionality can significantly reduce building energy consumption while maintaining year-round thermal comfort. The summer cooling performance of TAPSS film in Hong Kong was investigated using a custom-designed thermal setup and compared to PT film ($R_{sol}$ = 2.9%, $\varepsilon_{8-13}$ = 94.6%) and encapsulated PNDE hydrogel as a control (Fig. 6a). During a continuous 80-hour measurement period, the TAPSS film achieved an average temperature reduction of 17.4 °C compared to PT film during midday (between 10:00 and 14:00), with an average solar intensity ($I_{solar}$) of 900 W/m² (Fig. 6b). This cooling performance was also validated by infrared images (Supplementary Fig. 38). The temperature of PT film fluctuated more significantly owing to stronger heat convection with ambient air and more sensitive to solar irradiation. The TAPSS film maintained a temperature below the ambient air temperature at night, even at a high-humidity condition (RH > 80%), due to its high thermal emissivity, with an average sub-ambient temperature difference of 3.0 °C and a maximum sub-ambient temperature difference of 4.2 °C. The photothermal effect of the PTPCC layer accounted for the higher temperature of the TAPSS film than the PNDE hydrogel during daytime, as most of the sunlight that passed through the PNDE hydrogel was reflected by the foam substrate.

To further evaluate its energy-saving potential in building applications, we conducted global-scale energy consumption simulations using EnergyPlus on a representative four-story mid-rise apartment model, comparing three roof scenarios: TAPSS film, PT film, and radiative cooling (RC) ceramic, with an undecorated roof as control (Supplementary Fig. 39 and Table 6). Our study encompassed nineteen cities spanning diverse climate zones based on the American Society of Heating, Refrigerating and Air-Conditioning Engineers (ASHRAE) Standard[69] (Supplementary Table 7). Results indicate that the TAPSS film significantly reduced cooling energy consumption in regions with anti-/deicing demands, achieving annual cooling energy savings of 6.1

GJ (12.0%) in Beijing and 6.6 GJ (13.5%) in Denver compared to PT films (Fig. 6c). More importantly, its intelligent temperature-responsive capability effectively mitigated the heating penalty associated with RC ceramics, reducing heating energy consumption by an average of 16.6 GJ (3.9%) in cold-climate cities like Beijing, Denver, Shenyang, Harbin, and Fairbanks (Supplementary Fig. 40). Year-round simulations confirmed that TAPSS-equipped buildings consistently achieved the lowest energy consumption and energy-saving performance across all climate zones (Fig. 6d, Supplementary Figs. 41 and 42), demonstrating their global viability as an energy-efficient envelope solution.

## Discussion

In summary, we demonstrate a trilayer TAPSS film capable of dynamically switching between photothermal anti-/deicing and overheating prevention modes. By integrating superhydrophobicity, thermochromism, photothermal conversion, and phase-change energy storage into a unified structure, our design overcomes the key bottleneck of current photothermal anti-/deicing materials, delivering efficient ice mitigation in winter while preventing overheating in summer. The synergistic interplay of a highly transparent and superhydrophobic top layer, a freeze-resistant thermochromic hydrogel interlayer, and a photothermal phase-change base layer enables a solar absorptance of 92% in cold conditions, facilitating rapid ice melting (410 s) and 10-fold freezing delay (1115 s) under −20 °C, along with superior deicing and defrosting performance on building roofs, aircraft wings and wind turbine blades. Concurrently, the thermochromic interlayer extends optical modulation beyond the visible spectrum of TCMMs to the VIS-NIR range, endowing the TAPSS film with a 62% solar modulation ability and a -17 °C temperature reduction in hot weather. EnergyPlus simulations further confirm its potential for year-round building energy savings across all climate zones, highlighting its practicality for scalable applications. More importantly, the UV

shielding property and robust superhydrophobicity of the top layer ensure long-term durability, retaining performance even after 90 days of outdoor exposure, 20 freeze-thaw cycles, and 100 heating-cooling cycles. With its unparalleled combination of functionalities, this work not only proposes a framework for next-generation anti-icing technologies but also offers sustainable solutions and widespread prospects for seasonal thermal management. We anticipate that this proof-of-concept will inspire the design of adaptive photothermal materials, while substantial opportunities remain to further enhance mechanical robustness and long-term durability under complex operating conditions encountered in practical applications.

Despite the aforementioned merits, several limitations should be addressed in future studies, including the potential environmental concerns associated with fluorinated components, the lack of dynamic emissivity regulation, and the challenges related to large-scale fabrication. The fluorinated acrylate resin used in the MNTS film belongs to perfluoroalkyl and polyfluoroalkyl substances (PFAS), which raise potential toxicity and environmental issues. Therefore, exploring environmentally benign alternatives such as hydrocarbon-based low-surface-energy polymers (e.g., polydimethylsiloxane and alkylsilane) or bio-derived hydrophobic coatings (e.g., plant waxes) is highly desirable. Beyond material substitution, dynamically tuning mid-infrared emissivity in response to ambient temperature remains a critical challenge for balancing radiative cooling and anti-icing performance during nighttime. This can be realized using an infrared-transparent polyethylene substrate coated with a low-emissivity Ag nanowire layer to encapsulate hydrogel[70], thereby suppressing excessive cooling under low-temperature conditions. Another direction lies in integrating photothermal phase change materials with complementary active anti-icing strategies, such as low-power Joule heating or waste-heat recovery, to mitigate icing risks in the absence of solar irradiation. Although the present synthesis involves multiple steps, costly raw materials, and limited scalability, there is a clear route toward addressing these issues. The multilayer architecture could be simplified by dispersing selective photothermal nanomaterials (e.g., cesium tungsten bronze) within hydrogel matrices; scalability could be enhanced through spray-coating and roll-to-roll UV-NIL techniques; and costs could be reduced by employing more abundant and cost-effective precursors without compromising functionality.

## Methods

### Materials

N-isopropylacrylamide (NIPAM, >98%, TCI), N-methylolacrylamide (MBA, 99%, Aladdin), N,N,N′,N′-tetramethylethylenediamine (TEMED, 99%, Aladdin), ammonium persulfate (APS, 99.99%, Aladdin), ethylene glycol (EG, >99%, Aladdin), N,N-dimethylacrylamide (DMAA, >99.0 %, TCI), anodic aluminum oxide (AAO) template (Topmembranes Technology Co., Ltd, China), polydimethylsiloxane (PDMS Sylgard 184, Dow Corning Corporation), multiwalled carbon nanotubes (MWCNTs, >95%, inner diameter: 5-12 nm, outer diameter: 30–50 nm, length: 10–20 µm, Macklin), N-tridecane (>98%, Thermo Scientific), N-tetradecane (99%, Thermo Scientific), ethyl acetate (≥99.5%, Aladdin).

### Fabrication of MNTS film

The MNTS film was fabricated based on an ultra-thin AAO template supported by a polymeric adhesive coating. The AAO template featured nanoholes with a diameter of 60 nm, a pitch of 125 nm, and a depth of 200 nm. To create moth-eye nanostructures, the AAO mold was etched in an aqueous solution of copper chloride (CuCl$_2$) and hydrochloric acid (HCl) at a constant temperature of 55 °C for 1 h[71]. Subsequently, the etched mold was treated with a perfluoropolyether (PFPE, Solvay Solexis AF30, Nicca Korea Co.) releasing agent to enhance de-molding performance. The MNTS film was then produced via UV-NIL. Briefly, a UV-curable fluorinated acrylate (GD-TE-2206, GDNANO Co., Ltd., China) was dispensed onto the etched mold and

covered with a PET film, which provided rigidity while maintaining flexibility after curing. The UV resin was fully infused into the mold and cured under 0.1 MPa pressure with UV irradiation (wavelength: 365 nm, dose: 100 mW/cm²) applied to the PET film's backside for 60 s. After demolding, a flexible MNTS film was successfully obtained.

### Synthesis of PNDE hydrogel

The PNDE hydrogel was prepared via a one-pot process. In a typical procedure, 4.0 g NIPAM, 0.2 g DMAA, and 40 mg MBA were added into 20 g EG aqueous solution, followed by stirring for 20 min. Next, 45 mg APS and 16 µL TEMED were successively added and stirred for 10 min. The entire stirring was carried out at room temperature. The as-prepared hydrogels were labeled as PND$_x$E$_y$ hydrogels, where x represents the weight percentage of DMAA to NIPAM and y represents the weight percentage of EG to water. For optical performance characterization, the solution was injected into a mold comprised of two 0.5 mm-thick silica glasses with a high transmittance in the full wavelength of the solar spectrum, followed by placing it in a refrigerator maintained at 4 °C for 2 h to disperse the heat released during the free radical polymerization.

### Preparation of PTPCC

First, 0.12 g MWCNTs and a certain amount of PCM (a mixture of n-tridecane and n-tetradecane with a mass ratio of 1:3) were added to 2.0 g PDMS. The solution was then magnetically stirred and ultrasonically dispersed for 30 min. PCM not only plays a role in storing thermal energy but also serves as a diluter to ensure the uniform dispersion of MWCNTs in PDMS. After adding 0.2 g curing agent, the solution was vacuum degassed and poured into a mold and cured at 50 °C for 4 h. The as-prepared composite phase change material was labeled as CPCM-x, where x was the weight percentage of PCM to PDMS. To avoid the surface turning white after solidification, a photothermal film without adding PCM was coated on the CPCM through a similar procedure. The as-prepared photothermal film was labeled as PT-y, where y was the weight percentage of MWCNTs to PDMS.

### Integration of TAPSS film

The TAPSS film was integrated through the following steps. First, a 0.1 mm-thick polyethylene PET film was placed onto the prepolymer of PTPCC under controlled pressure. Prior to bonding, the PET substrate was treated by plasma to improve surface energy, and the ratio of curing agent was optimized to enhance the adhesion of PTPCC to the PET substrate. After complete curing, the edges between PET and PTPCC were sealed with UV-curing glue to further strengthen the interfacial bonding. This bonding approach avoids covering the internal areas of the layers, thus having negligible influence on their optical and thermal performance. Subsequently, a 0.07 mm-thick MNTS film and the PET film were positioned on the top and bottom, respectively, and bonded together using double-sided tape to form a mold for encapsulating PNDE hydrogels. The edges of the mold were further sealed with UV-curing glue to prevent leakage and improve adhesion. The thickness of the hydrogels was controlled by the thickness of the double-sided tape. Subsequently, the prepolymer solution of the hydrogels was injected into the mold, and the assembly was placed in a refrigerator maintained at 4 °C for 2 h to complete the polymerization reaction.

### Basic characterization

The chemical composition was characterized by FTIR spectroscopy (Thermo Fisher Nicolet iS5, USA) in attenuated total-reflection (ATR) mode. Spectra were recorded over the range of 400–4000 cm$^{-1}$ at a resolution of 4 cm$^{-1}$, with an average of 32 scans conducted at room temperature. X-ray photoelectron spectroscopy (XPS, Thermo Scientific K-Alpha, USA) was also employed for chemical composition analysis. Spectra were acquired in constant pass energy mode at 150 eV, with an energy step of 0.10 eV. All binding energies were calibrated

using the C 1s peak at 284.8 eV as the reference. Thermo Scientific™Avantage (version: 5.9931) was used for analyzing XPS spectra. The microscopic morphology was examined by a field emission scanning electron microscope (FESEM, Quanta 450 FEG, FEI). For hydrogel samples, the microscopic morphology in cold and hot states was preserved by immediate immersion in liquid nitrogen, followed by freeze-drying (LGJ-12A, Beijing Sihuan Qihang Technology Co., Ltd., China) for 72 h. The surface roughness was analyzed using an atomic force microscope (AFM, Bruker, Dimension Icon) with ScanAsyst air tips, scanning an area of 3 μm × 3 μm. The thermal conductivity was measured by the transient hot wire method using a thermal conductivity meter (TC3000E, Xi'an Xiaxi Electronic Technology Co., Ltd., China). The thermal stability was determined by thermogravimetric analysis (TGA, Q50, TA Instruments, USA) with a heating rate of 10 °C/min from room temperature to 400 °C under $N_2$ atmosphere. The peel strength between each layer was measured by an electronic universal testing machine (INSTRON 5967, USA) following ASTM D903. More details for the characterization of surface wettability, optical performance, phase change properties, and photothermal performance were presented in the Supplementary Methods.

### Passive anti-icing and photothermal deicing/defrosting tests

The experimental system for anti-/deicing tests is illustrated in Supplementary Fig. 32. The cooling module comprised a custom-made transparent acrylic cold chamber (17 cm × 13 cm × 10 cm) enclosed by a 6 cm-thick foam box for thermal insulation. Two fixed cooling blocks, connected to a low-temperature cooling circulating pump (DLSB-5/80, Shanghai Cancun Instrument Equipment Co., Ltd.) via circulation tubes, facilitated heat exchange between the coolant and the air inside the chamber to create a cold environment. Humidity was controlled using desiccants. A syringe was employed to dispense droplets onto the cold surface, and a xenon lamp (SciSun-300, Sciencetech) served as a solar simulator for photothermal de-icing tests. A high-speed camera (Phantom, Micro C110) recorded the droplet's freezing and melting processes from a side view, while T-type thermocouples (±0.5 °C uncertainty) monitored ambient and sample temperatures. During the anti-icing experiments, the ambient temperature was maintained at −20 ± 1 °C, while the RH was maintained at 20 ± 5%.

(1) Static anti-icing test. A 10 μL water droplet was gently deposited onto the sample surface when the surface temperature was stable, and a digital camera was triggered to record the freezing process. The anti-icing performance was assessed based on the freezing time, defined as the duration from the initiation of the high-speed camera to the complete freezing, characterized by the formation of a peach-like shape with a cusp apex.

(2) Dynamic anti-icing test. A 10 μL water droplet was dropped from 4.5 mm height onto a 3° inclined surface placed in a cold chamber, and the dynamic process was recorded by a high-speed camera.

(3) Photothermal deicing test. A 10 μL water droplet and an ice cube (1 cm × 1 cm × 2 cm) were placed in the cold chamber (−20 °C, 20%RH) for 1 h to complete freezing. Subsequently, the solar simulator with a light intensity of 1 kW/m² was activated, and the melting process was recorded by a high-speed camera.

(4) Photothermal defrosting test, A frost layer with a thickness of ~0.5 mm was first formed on the samples glued on the aircraft model and wind turbine blade model in a cold chamber (−15 °C, 80%RH). The humidity was controlled via a humidifier. Subsequently, the solar simulator with a light intensity of 1 kW/m² was activated, and the defrosting process was recorded by a digital camera.

### Field investigation

The cooling performance measurements were carried out during clear-sky conditions in Hong Kong (6–10 June 2025). Square samples measuring 5 cm × 5 cm were placed in shallow recesses on the top of expanded polystyrene boxes with dimensions of 30 cm × 30 cm × 30 cm. The experimental platform was elevated 50 cm above ground level and installed in an open, unobstructed area to guarantee continuous and uniform solar exposure during the measurements. The backside temperature of each sample was monitored using calibrated T-type thermocouples. Temperature signals were collected at 10 s intervals via a data acquisition system (NI9213, NI9201, and CDAQ-9174, National Instruments) operated through LabVIEW software. Concurrently, ambient environmental conditions, including air temperature, relative humidity, wind velocity, and solar irradiance, were logged every minute using a commercial meteorological station (YG-BX, Wuhan YIGU Chenyun Technology Co., Ltd.). The weather station was installed within a 2 m radius of the samples to minimize environmental discrepancies and ensure reliable measurements.

### EnergyPlus simulation

Building energy simulations were performed using EnergyPlus version 22.1.0 to evaluate the energy-saving potential of the specimens. We employed a validated 4-story mid-rise apartment building prototype (46 m × 17 m × 3 m per floor) developed by the Pacific Northwest National Laboratory of the U.S. Department of Energy (Supplementary Fig. 40), which serves as a widely recognized benchmark model for residential buildings. The model incorporates code-compliant energy efficiency standards, typical operation schedules, realistic occupancy patterns, and standard lighting/miscellaneous loads with conventional HVAC systems. In our simulations, the specimens were implemented as the outermost roof layer, with their optical properties specified in Supplementary Table 6. We assessed performance across representative global climate zones as detailed in Supplementary Table 7.

### Reporting summary

Further information on research design is available in the Nature Portfolio Reporting Summary linked to this article.

## Data availability

The data that support the findings of this study are available within the paper and Supplementary Information. Source data are provided with this paper.

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

## Acknowledgements

This work was supported by the Hong Kong Research Grant Council via the Research Fellow Scheme with the reference number RFS2425-1S06 (to C.Y.T.), and via the General Research Fund (GRF) account 11200923 (to C.Y.T.). The work described in this paper was financially supported by City University of Hong Kong for the project 'Fostering Innovation for Resilience and Sustainable Transformation' (FIRST), officially endorsed by the United Nations Educational, Scientific and Cultural Organization (UNESCO) under the International Decade of Sciences for Sustainable Development (IDSSD) (2024-2033) via the internal City University of Hong Kong account of 9610739 (to C.Y.T.). This work was also supported by the National Natural Science Foundation of China (52206068 to F.C.) and a donation for a research project grant at City University of Hong Kong from Pacific Enterprise Solutions Limited (9220166 to C.Y.T.).

## Author contributions

Conceptualization: J.D., F.C., C.Y.T.; Methodology: J.D., W.W., X.L., H.L., X.C., F.C.; Data Analysis: J.D., W.W., Y.F., J.T., F.C.; Investigation: J. D., W.W.; Supervision: C.Y.T.; Writing-original draft: J.D.; Writing-review and editing: J.D., F.C., Q.M., C.Y.T.

## Competing interests

The authors declare no competing interests.
