## [Transparent Peer Review file · Nature Communications]

A Self-Regulated Photothermal Anti-/Deicing Film for All-Season Applications

Corresponding Author: Professor Chi Yan TSO

Version 0:

Reviewer comments:

Reviewer #1

(Remarks to the Author)

The work of Du et al presents a multi-layered strategy for tackling ice accumulation on surfaces, combining superhydrophobicity with photothermal materials and thermochromic materials. To my knowledge, this is a novel strategy overall, even if each individual layer by itself is not overly innovative. I think the work is strong but requires revision before I would recommend publication. I also feel the authors have too many results in one single paper (especially in the SI), which actually detracts from the work rather than enhances it. They should consider which datasets are essential to their work and which perhaps belong in a future publication or a student thesis, rather than buried in the SI. Specific comments:

1. More information is needed on the fluorinated acrylate resin used, as it is likely a PFAS material that will be banned from worldwide usage in the next few years due to its environmental and human health toxicity. 41.6 at% F at the surface is quite a lot of hazardous chemical to release into the environment. The use of fluorinated coatings for superhydrophobicity no longer makes sense! Since the authors are not making commercial products, some discussion should be added on how the superhydrophobic layer of the TAPSS could be replaced with a PFAS-free alternative, in future works.
2. I would suggest the authors tone down their language for how 'amazing' their material system is. If the results are impressive, the readers will understand this. And some of the results are good, but not amazing. For example, withstanding 20 freeze-thaw cycles is good, but might occur over 10 days in a realistic environment whereas the coating should likely last many years.
3. The two proposed applications of wind turbine blades and airplane wings have substantial durability metrics, far beyond what was shown here. Moreover, weight is a major issue for both of those applications. The simulations on buildings also suggest that there are better applications for the TAPSS than aircraft/turbines. I would suggest refocusing the paper about static use cases like on roofs, rather than ones where more stringent durability is required. If the authors think differently, they should verify the durability of the TAPSS using metrics like rain erosion, sand erosion, and high-speed wind tunnel / icing wind tunnel testing.
4. I would like to see a part of the discussion devoted to the limitations of their TAPSS. Only the positives are discussed/presented, which is obviously not realistic.
5. How does contamination affect the performance of the TAPSS? Yes the superhydrophobic surface is self-cleaning for particles, but what about chemical contamination like oils or surfactants, both of which are present in real scenarios?
6. How scalable is the technology? The applications discussed all require many square meters of coated surface. Can AAO templating be achieved on such a scale?
7. The authors should characterize the inter-layer adhesion of the TAPSS between each of the layers. Hydrogels do not typically bond well to PDMS, for example.

Small comments:

- Pg 4 The use of the term "thermal comfort" should be reviewed throughout the manuscript. That typically refers to a human experience of comfort, which is not relevant to this work.
- Figure 1f: are these single references that are being compared to the TAPSS film, or several different papers? Regardless, citations are missing for these.
- End of page 8: relating high IR adsorption to FTIR results is incorrect as it is the magnitude of the absorption that is important, not the presence of the bonds.
- English writing throughout could be improved for clarity, focusing on simplifying word choice and sentence structure

Reviewer #2

(Remarks to the Author)

The paper proposes a design of a three-layer membrane structure, which features surface hydrophobicity, high-temperature radiative heat dissipation performance, and low-temperature solar absorption performance, with anti-icing as its application scenario, and demonstrates its application value. The design idea of the paper's work has certain innovativeness, but some key issues need further demonstration and explanation:

1 Phase change materials are used for heat storage to delay the icing time, and the delay time is related to the thickness of the phase change material coating. The thickness of the phase change coating used in the paper is limited, so the icing delay time in actual application scenarios is very short, which can hardly achieve practical anti-icing effects during long winter nights. How does the author consider this?

2 The hydrophilic-hydrophobic property change of the hydrogel is used to realize the switching between light reflection and light transmission. In the hydrophobic state, how to prevent the loss of released water? After water release, the volume of the hydrogel polymer will shrink; how to prevent or handle the impact of polymer deformation?

3 How are the three layers combined, and what is the bonding force? How is the overall structure of the three-layer structure encapsulated? In addition to the possible loss of water when the hydrogel is hydrophobic, the phase change material will also flow away after melting, which are related to the overall encapsulation of the structure.

4 The surface material has good radiative performance, which can exchange heat with outer space to reduce the material's temperature in summer. However, radiative cooling materials without temperature self-adaptive function still have cooling capacity at low temperatures in winter, especially during winter nights, radiative cooling will further lower the material's temperature, increasing the risk of icing. How to overcome this?

5 The material preparation process is complex, and the price of raw materials is high. What is the prospect of large-scale preparation of the material?

Reviewer #3

(Remarks to the Author)

This paper presents a development of a temperature-adaptive photothermal storage superhydrophobic film with a trilayer design, comprising a photothermal phase-change base layer, a freeze-resistant thermochromic hydrogel interlayer and a transparent superhydrophobic top layer.

This study is expected to provide a useful wearable large area rapid thermal management and this can be recommended for publication if the authors can successfully respond to the following comments,

1) Diverging from conventional materials with constrained functionality, the suggested design combines a broadband high-absorbance photothermal phase-change base layer with a freeze-resistant thermochromic hydrogel interlayer that exhibits thermal-comfort transition temperature and strong visible-near-infrared (VIS-NIR) modulation capability. The integration between tri-layers will need integration. The integration layer for attaching with some chemicals will affect the optical and thermal performance between layers. This should be briefly discussed in the manuscript because it will affect the resultant performance.

2) The photothermal performance of PTPCC base layer determines the deicing efficiency of TAPSS film, which was optimized by tuning MWCNTs loading. At 2 wt% MWCNTs content, a broadband solar absorbance of nearly 97% was achieved (Fig. 4a) due to continuous energy levels of hybrid bonds. It is well understood the MWCNT will strongly absorb the sunlight. However, I wonder how the sunlight will be absorbed for the non-transparent mode. I wonder if the hydrogel absorbs the sunlight or hydrophobic layer absorb sunlight. Even though they may absorb sunlight, it is expected to be weak.

3) Continuing the previous comment, the general transient temperature measurement starting from the non-transparent mode needs to be provided. I guess the temperature evolution in Figure 4(i) is for the case right after the time when the bottom light absorbing layer is exposed. I wonder how long it takes for the transition from non-transparent to transparent mode transition as soon as the sunlight start to irradiate (not just a cyclic test shown in figure 6(b)).

4) The adaptive functionality of TAPSS film necessitates a thermochromic interlayer that exhibits high transparency in the cold state, strong solar modulation capability and a suitable transition temperature. Among various explored thermochromic materials including vanadium dioxide (ACS Appl. Mater. Interfaces 13, 22495-22504 (2021)) and perovskite (Adv. Funct. Mater. 31, 2010426 (2021)), thermochromic hydrogels, particularly poly(N-isopropylacrylamide) (PNIPAM)-based systems (J. Mater. Chem. A 2, 13550-13555 (2014)), outperform due to superior solar modulation ability and thermal responsiveness. Recent related progresses in the optical transparency change by phase change material (Sci. Adv., 5, eaav4916, (2019)), dynamic crosslinking of liquid crystal elastomer (Nat. Mater., 23, 834 (2024)) need to be briefly discussed in the introduction part to provide the readers with the recent related progresses.

5) Figure 1(e) presents the absorption spectra of the TAPSS film in heating and cooling modes. There are some issues to be cleared for this optical property measurement. The adaptive functionality of TAPSS film necessitates a thermochromic interlayer that exhibits high transparency in the cold state, strong solar modulation capability and a suitable transition temperature. I wonder if the optical property was measured with the ice formed on the surface because ice is basically water and water molecule will be strongly absorbed at some peaks in NIR wavelength.

6) Continuing the previous comments, there seems to be still strong absorption over 1 micron wavelength even for the cooling mode. This may mean considerable amount of solar energy will be absorbed even for the cooling mode and it may induce an unexpected mode transition for the prolonged use under a certain condition. I wonder how this can be overcome.

7) Unlike conventional photothermal materials, our TAPSS film not only enables efficient anti-/deicing in winter but also prevents overheating in summer due to its exceptional solar regulation capability. This dual functionality can significantly reduce building energy consumption while maintaining year-round thermal comfort. I wonder where the cooling functionality comes from. Is it just reflection effect or is there any radiative cooling effect?

8) The authors proposed a temperature-adaptive photothermal storage superhydrophobic film featuring a rationally engineered multilayer structure by separating solar absorption layer and light modulation layer etc. Similar approach has been applied to the tunable radiative cooling (Small, 20, 2308572 (2024); Nanophotonics, 13, 543 (2024)) and to maximize photothermal absorption on the multifunctional multilayered structures (Joule, 9, 102007 (2025); Small, 20, 2304338 (2024)) for radiative cooling. They need to be briefly discussed in the introduction part to provide the readers with the recent related progresses.

9) To preserve the photothermal performance of the underlying layer while delaying ice formation, the top layer must simultaneously achieve high transparency and superhydrophobicity, which was fabricated via ultraviolet nanoimprint lithography (UV-NIL, Supplementary Fig. 1) to precisely engineer subwavelength nanostructures. Nanoimprint lithography usually suffers from the large area production yet. I wonder what is the largest size can be produced now and what is the limiting factor and how can it be overcome?

10) I wonder if the same sample shows different performance depending on the thermal property of the surface where the sample is mounted, for example, on the wood or on the metal surface.

11) Some of the supplementary movies are not real time. The playing speed should be indicated on the whole movie clips (not just the title pages).

12) The general usage of language including typos and grammatical errors need to be checked again.

13) Some of the digital pictures are missing scale bars. They should be added to the pictures.

Reviewer #4

(Remarks to the Author)

This manuscript presents a temperature-adaptive photothermal storage superhydrophobic (TAPSS) film designed to mitigate the overheating issue commonly observed in traditional photothermal surfaces during hot seasons. The TAPSS film features an innovative trilayer structure, comprising a photothermal phase-change base layer, a freeze-resistant thermochromic hydrogel interlayer, and a transparent superhydrophobic top layer. This design enables the film to exhibit excellent anti-/deicing performance under cold conditions, along with remarkable radiative cooling performance in hot environments. Additionally, the film demonstrates strong UV resistance and mechanical durability, highlighting its potential for practical applications in diverse scenarios. Overall, the manuscript is innovative and supported by comprehensive experimental data, representing a notable advancement in the field of intelligent anti-/deicing technology. It is recommended for publication after addressing the following issues.

1. According to Fig. 1e, the solar modulation ability decreases significantly at wavelengths beyond 1.2 μm . What is the underlying reason for this reduction? Could the authors propose potential solutions, such as employing alternative thermochromic hydrogels, to enhance the solar modulation performance in this spectral range?

2. The authors used SEM images and pore size distribution data to explain that the opacity of thermochromic hydrogels above the LCST arises from enhanced light scattering. It would be beneficial to include additional quantitative evidence, such as light scattering efficiency simulations, to further substantiate this mechanism.

3. In Fig. 3a and 3b, the enthalpy peak associated with the phase change process appears to diminish with the addition of ethylene glycol. What explains this reduction in phase change enthalpy?

4. In Fig. 4f, the cooling rates of CPCM-80 and PTPCC are quite similar, yet the heating rate of CPCM-80 is considerably slower than that of PTPCC. Could the authors elaborate on the mechanism responsible for this difference?

5. The deicing time of the TAPSS film is somewhat longer compared to other photothermal surfaces reported in the literature, which may be attributed to the thermal resistance of the upper layers and the thermal energy storage behavior of the bottom layer. What strategies could be proposed to accelerate the ice-melting process?

6. How is the adhesion between the bottom and middle layers ensured to prevent delamination during operation? Are there specific interfacial treatments or material modifications employed to enhance layer integration?

Version 1:

Reviewer comments:

Reviewer #1

(Remarks to the Author)

I thank the authors for diligently responding to all of my comments and concerns. In general I think the manuscript is greatly improved, and would be suitable for publication. Two small comments that the authors may wish to consider prior to final publication:

1. The tetradecane immersion test is poorly described in your methods and not really a great demonstration of resistance to contamination, as I assume the oil was washed off prior to testing the subsequent superhydrophobicity? Since the surface was clearly wet by tetradecane it indicates the surface is not superoleophobic, which is ok as that was not the goal of your study, but again is a limitation that could be recognized rather than hid / downplayed. This is also another reason using PFAS is not required, as typically the only reason PFAS is necessary is to achieve things like superoleophobicity.

2. There is still a disconnect between all the applications / uses that the authors claim the coating will be good for, and the more practical discussion given in the Response file. For example, if the coating is just a demonstration and proof of concept, it will not potentially find any real world use or have any potential benefit to roofs, aircraft, etc. Right now the authors

are trying to 'have their cake and eat it too', ie claim that the coating will indeed save the world, but also that's it's just a proof of concept and so all the shortcomings can be ignored.

Reviewer #2

(Remarks to the Author)

The authors have provided detailed and reasonable answers and responses to the questions raised, and the paper can now be considered for publication.

Reviewer #3

(Remarks to the Author)

The authors responded well to the comments.

Reviewer #4

(Remarks to the Author)

The response from authors is satisfactory and I recommend publishing this work now.

Response Letter

Four reviewers have responded to our manuscript and provided very conducive comments. We really appreciate their efforts and time on our manuscript. We have taken all the reviewers' suggestions, clarified their queries, and carefully revised the manuscript. In the revised manuscript, all changes are highlighted in blue for easy review, and a list of main changes made is summarized as follows.

In Manuscript:

1. Added a discussion of the limitations (e.g., environmental concern, scalability, fixed emissivity, etc.) of temperature-adaptive photothermal storage superhydrophobic (TAPSS) film, along with potential strategies to address these issues.
2. Added acid-rain impact and sand impact tests to evaluate the durability of transparent superhydrophobic films.
3. Added photothermal deicing experiments demonstrating the performance of TAPSS film when applied to building roofs and the results were added in Figure 5d.
4. Added oil and surfactant repellency tests, as well as corresponding durability assessments, for the transparent superhydrophobic film.
5. Added peel strength tests between each layer within the TAPSS film, along with detailed explanations of the adhesion-enhancement and encapsulation strategies.
6. Added references to recent advances in thermochromic and adaptive materials.
7. Simplified wording and improved sentence structures throughout the manuscript for better clarity and readability.

In Supporting Information:

1. Added Supplementary Figure 7, Figure 10, Figure 16.
2. Removed Supplementary Figures 12, 29, 34 and 37 in the original SI and adjusted the order of the following figures
3. Added detailed experimental procedures for characterizing the durability of superhydrophobicity and the peel strength between layers.

Reviewer #1:

General comments: The work of Du et al presents a multi-layered strategy for tackling ice accumulation on surfaces, combining superhydrophobicity with photothermal materials and thermochromic materials. To my knowledge, this is a novel strategy overall, even if each individual layer by itself is not overly innovative. I think the work is strong but requires revision before I would recommend publication. I also feel the authors have too many results in one single paper (especially in the SI), which actually detracts from the work rather than enhances it. They should consider which datasets are essential to their work and which perhaps belong in a future publication or a student thesis, rather than buried in the SI.

General responses: We sincerely appreciate the reviewer's positive assessment of our work and the constructive suggestions provided. Following the reviewer's advice, we have carefully re-evaluated the datasets included in the Supplementary Information (SI) and removed figures that are not essential to supporting the main findings (i.e., Figs. 12, 29, 34 and 37 in the original SI). The corresponding results will be further refined and presented in our future studies or student theses. We have thoroughly revised the manuscript to improve clarity and focus, and have addressed all other comments in detail as follows:

Comment 1: More information is needed on the fluorinated acrylate resin used, as it is likely a PFAS material that will be banned from worldwide usage in the next few years due to its environmental and human health toxicity. 41.6 at% F at the surface is quite a lot of hazardous chemical to release into the environment. The use of fluorinated coatings for superhydrophobicity no longer makes sense! Since the authors are not making commercial products, some discussion should be added on how the superhydrophobic layer of the TAPSS could be replaced with a PFAS-free alternative, in future works.

Response 1: We thank the reviewer for raising this important concern regarding the environmental and health risks associated with fluorinated acrylate resins. We fully acknowledge that the fluorinated acrylate resin used in our work, 1H,1H,2H,2H-perfluorodecyl acrylate, is a PFAS-type material, and that the global use of PFAS-containing polymers is increasingly restricted due to their persistence and potential toxicity. In this study, the fluorinated resin was employed solely for proof-of-concept demonstration to achieve superhydrophobicity within the TAPSS structure, rather than for any direct commercialization purpose. To address this concern, we have revised the manuscript to explicitly clarify the limited, research-only use of PFAS-containing materials and have added a discussion on

potential PFAS-free alternatives.

In particular, future work could explore environmentally friendly, PFAS-free strategies such as: (a) Hydrocarbon-based low-surface-energy polymers, e.g., polydimethylsiloxane (Applied Surface Science, 2015, 339, 94 – 101) and alkylsilane coatings (ACS Omega, 2019, 4(4), 6947 – 6954); (b) Bio-based hydrophobic coatings, such as plant wax – derived materials (Science, 2012, 335(6064), 67 – 70). These alternative routes provide sustainable approaches to replace fluorinated coatings while maintaining the desired superhydrophobic functionality of TAPSS. The corresponding discussion has been added to the revised manuscript and marked in blue (see Discussion section on page 20), and the chemical structure of the fluorinated acrylate resin has been added to Supplementary Fig. 1.

Fig. R1 (Supplementary Fig. 1). Fabrication procedures of moth-eye nanostructured transparent superhydrophobic (MNTS) film.

[Revised contents in the revised Manuscript]

Page 20: *The fluorinated acrylate resin used in the MNTS film belongs to perfluoroalkyl and polyfluoroalkyl substances (PFAS), which raise potential toxicity and environmental issues. Therefore, exploring environmentally benign alternatives such as hydrocarbon-based low-surface-energy polymers (e.g., polydimethylsiloxane and alkylsilane) or bio-derived hydrophobic coatings (e.g., plant waxes) is highly desirable.*

Comment 2: I would suggest the authors tone down their language for how ‘amazing’ their material system is. If the results are impressive, the readers will understand this. And some of the results are good, but not amazing. For example, withstanding 20 freeze-thaw cycles is good, but might occur over 10 days in a realistic environment whereas the coating should likely last many years.

Response 2: We thank the reviewer for this valuable suggestion. We agree that the language in the original manuscript may have been overly enthusiastic in describing the performance of our material.

In the revised version, we have carefully toned down the wording to present the results in a more balanced and objective manner. Specifically, we reduce overly flowery words like ‘exceptional’, ‘remarkable’, ‘outstanding’, ‘superior’.

For example, instead of describing the freeze–thaw durability as ‘exceptional’, we now describe it as ‘good’ while we have also added clarification that withstanding 20 freeze–thaw cycles demonstrate initial durability but do not fully reflect the requirements for long-term service lifetimes, which can extend over many years in practical environments. **This limitation is now explicitly discussed in Anti-/deicing and defrosting performance section (Marked in blue on page 17).**

[Revised contents in the revised Manuscript]

Page 17: *Notably, the TAPSS film retained its excellent superhydrophobicity and optical properties after 20 freeze-thaw cycles (Supplementary Fig. 37). Although these results demonstrate promising initial durability, further improvement is necessary to meet the long-term service requirements for practical applications, which may span several years.*

Comment 3: The two proposed applications of wind turbine blades and airplane wings have substantial durability metrics, far beyond what was shown here. Moreover, weight is a major issue for both of those applications. The simulations on buildings also suggest that there are better applications for the TAPSS than aircraft/turbines. I would suggest refocusing the paper about static use cases like on roofs, rather than ones where more stringent durability is required. If the authors think differently, they should verify the durability of the TAPSS using metrics like rain erosion, sand erosion, and high-speed wind tunnel/icing wind tunnel testing.

Response 3: We appreciate the reviewer’s insightful comments. We fully agree that aircraft wings and wind turbine blades impose stringent requirements on material weight, mechanical strength, and long-term durability, which our current TAPSS film may not yet meet. In our study, the demonstrations involving these scenarios were intended to illustrate the potential of TAPSS film for anti-/deicing functionality rather than to claim immediate applicability to such demanding environments. We agree that, at the present stage, the TAPSS film is more suitable for static use cases such as building roofs or outdoor infrastructures. Accordingly, we supplemented our work with additional deicing tests on the roof surface of a house model. The results show that on the TAPSS-coated side, the ice cube melted and slid downward under gravity at 555 s, whereas on the uncoated

side, the ice cube remained unmelted and firmly adhered to the surface. The corresponding results are presented in Fig. 5d and discussed in the revised manuscript (see pages 17 and 20, marked in blue).

Fig. R2 (Fig. 5d). Deicing process of ice cubes (1 cm×1 cm×2 cm) on the roof surfaces of a model house with one side coated with TAPSS film under 1.0 sun illumination (−20 °C, 20% RH).

Furthermore, following the reviewer’s suggestions, we conducted additional durability evaluations, including acid rain corrosion and sand impact resistance tests. The results show that the superhydrophobicity can maintain stable performance after **2 hours of acid rain exposure and impact from 1000 g of sand particles**, demonstrating good chemical and mechanical robustness suitable for long-term outdoor operation. The detailed experimental procedures have been added to **Supplementary Information** (see page 3), and the corresponding results are presented in **Supplementary Fig. 16** and discussed in the revised manuscript (see pages 9 and 18, marked in blue).

Fig. R3 (Supplementary Fig. 16). Changes in CA and SA of MNTS film during **a** sand impact and **b** acid rain impact tests.

[Revised contents in the revised Manuscript]

Page 9: *Our film also withstood continuous manual abrasion (Supplementary Movie 4), high-pressure water jets (Supplementary Movie 5), sand impact of 1000 g and acid rain exposure for 2 hours (Supplementary Fig. 16)*

Page 17: *The deicing performance was further evaluated using ice cubes placed on the roof surface of a house model. On the TAPSS-coated side, the ice cube melted and slid downward under gravity at 555 s, whereas on the uncoated side, the ice cube remained unmelted and firmly adhered to the surface (Fig. 5d).*

Page 18: *Durability tests of the TAPSS film under acid rain and sand impact (Supplementary Fig. 16) further confirmed its potential for dynamic applications such as wind turbine blades and aircraft wings.*

Page 20: *along with superior deicing and defrosting performance on building roofs, aircraft wings and wind turbine blades*

[Revised contents in the revised Supporting Information]

Page 3: *(3) Acid rain impact test. We used a shower with a nozzle diameter of 0.4 mm to form simulated acid rainfall (pH=5.0) with a flow of ~20 mL/s. The 15° tilted sample was placed 30 cm beneath the shower. The CA and SA were measured after certain time intervals.*

(4) Sand impact test (ASTM D968). The silica sand (200-300 μm) fell freely from a height of 15 cm to impact the surface at a tilt angle of 45°. The CA and SA were measured after falling of certain mass of sand particles.

Comment 4: I would like to see a part of the discussion devoted to the limitations of their TAPSS. Only the positives are discussed/presented, which is obviously not realistic.

Response 4: We appreciate the reviewer's constructive suggestion. We agree that discussing the limitations of the TAPSS film is essential for providing a balanced perspective. In the revised manuscript, we have added a new paragraph in the Discussion section to address the current limitations of the TAPSS film, including the use of fluorinated components with potential environmental concerns, the lack of dynamic emissivity regulation, the limited anti-icing duration during nighttime, and the challenges for large-scale fabrication, etc. Corresponding strategies and potential solutions for overcoming these issues have also been proposed in the revised manuscript and marked in blue (see Discussion section on pages 20 and 21).

[Revised contents in the revised Manuscript]

Page 20 and 21: *Despite the aforementioned merits, several limitations should be addressed in future studies, including the potential environmental concerns associated with fluorinated components, the*

lack of dynamic emissivity regulation, and the challenges related to large-scale fabrication. The fluorinated acrylate resin used in the MNTS film belongs to perfluoroalkyl and polyfluoroalkyl substances (PFAS), which raise potential toxicity and environmental issues. Therefore, exploring environmentally benign alternatives such as hydrocarbon-based low-surface-energy polymers (e.g., polydimethylsiloxane and alkylsilane) or bio-derived hydrophobic coatings (e.g., plant waxes) is highly desirable. Beyond material substitution, dynamically tuning mid-infrared emissivity in response to ambient temperature remains a critical challenge for balancing radiative cooling and anti-icing performance during nighttime. This can be realized using an infrared-transparent polyethylene substrate coated with a low-emissivity Ag nanowire layer to encapsulate hydrogel, thereby suppressing excessive cooling under low-temperature conditions. Another direction lies in integrating photothermal phase change materials with complementary active anti-icing strategies, such as low-power Joule heating or waste-heat recovery, to mitigate icing risks in the absence of solar irradiation. Although the present synthesis involves multiple steps, costly raw materials, and limited scalability, there is a clear route toward addressing these issues. The multilayer architecture could be simplified by dispersing selective photothermal nanomaterials (e.g., cesium tungsten bronze) within hydrogel matrices; scalability could be enhanced through spray-coating and roll-to-roll UV-NIL techniques; and costs could be reduced by employing more abundant and cost-effective precursors without compromising functionality.

Comment 5: How does contamination affect the performance of the TAPSS? Yes the superhydrophobic surface is self-cleaning for particles, but what about chemical contamination like oils or surfactants, both of which are present in real scenarios?

Response 5: We appreciate the reviewer's insightful comment regarding the effect of chemical contamination on the performance of the TAPSS. We agree that, although the superhydrophobic top layer exhibits excellent self-cleaning ability toward solid particles, low-surface-tension organic contaminants or surfactants may reduce the water contact angle and increase the sliding angle, thereby weakening the anti-icing performance. To address this concern, we evaluated the repellency of the superhydrophobic surface to both oily and surfactant contamination. The results show that the surface maintains stable superhydrophobicity when exposed to sodium dodecyl sulfate (SDS) surfactant solutions below 0.5 times the critical micelle concentration (CMC), whereas its oleophobicity to

tetradecane is relatively weaker due to low surface tension (~26 mN/m). After immersion in tetradecane for 120 hours, the surface still exhibited negligible degradation in hydrophobicity. These results suggest that in practical applications, even when chemical contaminants are present, the surface can restore its superhydrophobicity after natural rinsing by rainwater. According to your suggestion, corresponding results and analysis were provided in the revised manuscript and marked in blue (see page 8) and Supplementary Fig. 10.

Fig. R4 (Supplementary Fig. 10). **a** Changes in CA and SA of sodium dodecyl sulfate (SDS) aqueous solution droplets with different concentration. **b** CA of a tetradecane droplet on the MNTS film. **c** Changes in CA and SA of water droplets after immersion in tetradecane.

[Revised contents in the revised Manuscript]

Page 8: *In addition to particles, the chemical contamination resistance was further evaluated using sodium dodecyl sulfate (SDS) and tetradecane. The surface maintained a high CA (>150°) and low SA (<10°) when the SDS concentration was below 0.5 critical micelle concentration (CMC) (Supplementary Fig. 10). Although the surface showed weaker oleophobicity toward tetradecane (CA~95°), the rinsed surface remained stable superhydrophobicity after immersion in tetradecane for 120 h. These results confirm that our MNTS film possesses good contamination robustness.*

Comment 6: How scalable is the technology? The applications discussed all require many square meters of coated surface. Can AAO templating be achieved on such a scale?

Response 6: We thank the reviewer for raising this important question on scalability. The largest scale we can currently achieve is **10 cm*10 cm** (see Fig. 1b). We acknowledge that while anodic aluminum oxide (AAO) templating is a well-established laboratory technique capable of producing highly

ordered nanostructures, extending this scale to many square meters is indeed challenging due to limitations in electrolyte management, heat dissipation, and membrane integrity. To clarify this point, we have revised the manuscript to note that the current work primarily demonstrates the feasibility of TAPSS fabrication at the laboratory scale. For large-area applications such as building roofs or aircraft wings, we envision scalable strategies by employing **roll-to-roll nanoimprint lithography** (Journal of Vacuum Science & Technology B, 2015, 33(6)). This approach is more practical for producing coatings over many square meters while retaining the required surface functionality. This discussion has now been added to the revised manuscript and marked in blue (see Discussion section on page 21).

[Revised contents in the revised Manuscript]

Page 21: *scalability could be enhanced through spray-coating and roll-to-roll UV-NIL techniques; and costs could be reduced by employing more abundant and cost-effective precursors without compromising functionality.*

Comment 7: The authors should characterize the inter-layer adhesion of the TAPSS between each of the layers. Hydrogels do not typically bond well to PDMS, for example.

Response 7: We thank the reviewer for this valuable comment. To address this concern, peel tests (ASTM D903) were conducted to characterize the interlayer adhesion within the TAPSS structure. The results confirmed strong bonding between the layers, with maximum peel strengths of $384 \text{ N}\cdot\text{m}^{-1}$ for the top two layers and $78 \text{ N}\cdot\text{m}^{-1}$ for the bottom two layers. Compared to reported peel strengths of PDMS-based composite coatings, our peel strength is higher. For example, $\sim 19 \text{ N/m}$ for skin-adhesive GCD-PDMS films (Science Translational Medicine, 2023, 15(693): eabq1634) and $\sim 60 \text{ N/m}$ for CNT/PDMS conductive elastomer electrodes (Nanoscale, 2025, 17(14): 8624-8633). It should be noted that the hydrogel layer is not directly attached to the PDMS layer; instead, it is encapsulated between the transparent superhydrophobic top layer and a PET film. The hydrogel edges were sealed with double-sided adhesive tape and UV-curable glue to enhance interfacial bonding and prevent leakage. On the other side, the PDMS layer was cured onto a plasma-treated PET substrate to improve surface adhesion, and their edges were similarly reinforced with UV-curable glue. Following the reviewer's suggestion, we have added more detailed descriptions of the layer integration and bonding procedures in pages 7 and 23 (marked in blue) of the revised manuscript and in Supplementary Fig.

7.

Fig. R5 (Supplementary Fig. 7). Peel force versus displacement for MNTS film-PNDE hydrogel and PET film-PTPCC interfaces.

[Revised contents in the revised Manuscript]

Page 7: *To ensure robust interfacial bonding among the multilayer components, several strategies were employed. The PET substrate, which serves both as the encapsulation layer for the hydrogels and the bonding interface with the PTPCC layer, was plasma-treated to promote adhesion. The curing agent ratio of PDMS was further optimized to strengthen its bonding with the PET substrate. Additionally, adhesive tape and UV-curable glue was applied along the edges to reinforce adhesion and prevent hydrogel leakage. Peel tests confirmed that the MNTS film and PNDE hydrogel were effectively bonded with a maximum peel strength of 384 N/m, while the adhesion between the PET film and PTPCC layer reached 78 N/m (Supplementary Fig. 7).*

Page 23: *Prior to bonding, the PET substrate was treated by plasma to improve surface energy, and the ratio of curing agent was optimized to enhance the adhesion of PTPCC to the PET substrate. After complete curing, the edges between PET and PTPCC were sealed with UV-curing glue to further strengthen the interfacial bonding. This bonding approach avoids covering the internal areas of the layers, thus having negligible influence on their optical and thermal performance. Subsequently, a 0.07 mm-thick MNTS film and the PET film were positioned on the top and bottom, respectively, and bonded together using double-sided tape to form a mold for encapsulating PNDE hydrogels. The edges of the mold were further sealed with UV-curing glue to prevent leakage and improve adhesion.*

Comment 8: The use of the term “thermal comfort” should be reviewed throughout the manuscript. That typically refers to a human experience of comfort, which is not relevant to this work.

Response 8: We appreciate the reviewer’s insightful comment regarding the use of the term ‘thermal comfort’. We agree that, for applications such as aircraft, wind turbines, and transmission lines, human comfort is not a relevant consideration. In these cases, only the upper temperature limits are typically specified. For instance, wind turbine surfaces should remain below 80-120 ° C (Renew. Sustain. Energy Rev. 208, 114983), high-voltage transmission lines below 90 ° C (Int. J. Fatigue 188, 108515), and photovoltaic modules below 70 ° C to avoid a ~20% efficiency loss (Appl. Energy 394, 126190). However, setting a transition temperature in such a high temperature range would not serve any effective cooling purpose. In contrast, for building applications, there exists an optimal temperature range associated with human thermal comfort, which we used as a reference to assess whether the hydrogel’s transition temperature is appropriate. This transition temperature is far below the failure thresholds of the aforementioned devices, providing an additional cooling protection effect. Thus, we believe that this value is also beneficial for equipment such as aircraft, wind turbines, and power lines. Moreover, the transition temperature of our thermo-responsive hydrogel is highly tunable. It can be increased by raising the DMAA content or decreased by adding more ethylene glycol, allowing us to tailor it to different application scenarios. According to the reviewer’s suggestion, we have clarified this point in the revised manuscript on page 6 and replace some ‘thermal comfort’ with ‘suitable’ on pages 4 and 11 (marked in blue).

Comment 9: Figure 1f: are these single references that are being compared to the TAPSS film, or several different papers? Regardless, citations are missing for these.

Response 9: We appreciate the reviewer’s careful observation. The data shown in Figure 1f are the averaged values summarized in Supplementary Table 1, which were collected from multiple literature sources. Because the references are numerous and already listed in Supplementary Information, we have added a note in the caption of Fig. 1f (marked in blue) to clarify that the data were obtained from Supplementary Table 1, rather than citing all references individually in the figure.

Comment 10: End of page 8: relating high IR adsorption to FTIR results is incorrect as it is the magnitude of the absorption that is important, not the presence of the bonds.

Response 10: We thank the reviewer for highlighting this important point. We fully agree that the infrared (IR) absorption and emissivity depend primarily on the magnitude of absorption rather than merely the presence of specific chemical bonds. In our MNTS film, the broadband high emissivity cannot be solely attributed to the strong C–O bond, as it does not cover the entire atmospheric window. Instead, the overall emissivity arises from the collective and cumulative vibrational contributions of multiple functional groups, including C–O, C–H, and C–F bonds, whose overlapping absorption bands enhance the overall IR absorption magnitude across the 8–13 μm range (Nature Communications, 2021, 12, 365; Materials Today Physics, 2019, 10, 100127). We have revised the corresponding discussion in the revised manuscript on page 9 (marked in blue) to clarify this point.

[Revised contents in the revised Manuscript]

Page 9: *This high value arose from the collective and cumulative effects of C–O stretching vibrations, C–H bending vibrations, C–F stretching and bending vibrations in the range of 770–1250 cm^{-1}*

Comment 11: English writing throughout could be improved for clarity, focusing on simplifying word choice and sentence structure.

Response 11: We thank the reviewer for this valuable suggestion. The entire manuscript has been carefully revised to improve clarity and readability. Word choice and sentence structures have been simplified to enhance fluency and ensure that the scientific ideas are conveyed in a clear and concise manner. Specifically, we make the following modifications, which were marked in blue in the revised manuscript.

(1) **On page 3, original sentence:** ‘researchers further developed photothermal storage superhydrophobic materials (PSSMs) that incorporate thermal energy storage capabilities of phase change materials (PCMs), allowing daytime solar energy accumulation and subsequent latent heat release for nighttime ice prevention.’

Simplified sentence: photothermal storage superhydrophobic materials (PSSMs) incorporating phase-change materials were developed to store daytime solar heat and release it at night for ice prevention.

(2) **On page 4, original sentence:** ‘the operational temperature of wind turbines needs to remain below 80–120 $^{\circ}\text{C}$ to avoid thermal stress and expansion, otherwise their generator lifespan and rated capacity will be compromised’

Simplified sentence: ‘wind turbines must operate below 80-120 °C to prevent thermal stress that shortens generator lifespan and rated capacity’

(3) **On page 8, original sentence:** ‘Coating polyethylene terephthalate (PET) substrates with MNTS film achieved high transmittance ($\tau > 95\%$) in the 500-1000 nm range, which was a substantial improvement over uncoated PET ($\tau \sim 88\%$, Fig. 2c). This optical enhancement originated from biomimetic antireflective moth-eye nanostructures that reduced solar reflectivity from 11.8% to 4.3% (Supplementary Fig. 9).’

Simplified sentence: MNTS-coated polyethylene terephthalate (PET) showed much higher transmittance ($\tau > 95\%$) than bare PET ($\tau \sim 88\%$, Fig. 2c) due to biomimetic moth-eye nanostructures that lowered reflectivity from 11.8% to 4.3%.

(4) **On page 11, original sentence:** ‘Although this modification caused minor reductions in $R_{\text{sol},40^\circ\text{C}}$ (from 77.0% to 74.4%) and reflectivity modulation (ΔR_{sol} , from 70.4% to 67.8%), it preserved satisfying $\tau_{\text{sol},20^\circ\text{C}}$ ($>82\%$, Fig. 3d and e).’

Simplified sentence: ‘Although this modification caused minor reductions in $R_{\text{sol},40^\circ\text{C}}$ and reflectivity modulation (ΔR_{sol}), it preserved satisfying $\tau_{\text{sol},20^\circ\text{C}}$ ($>82\%$, Fig. 3d and e).’

(5) **On page 20, original sentence:** ‘With its unparalleled combination of functionalities, this work not only proposes a new framework for next-generation anti-icing technologies through collaborative innovation of materials, structure and function’

Simplified sentence: ‘With its unparalleled combination of materials and functionalities, this work not only proposes a new framework for next-generation anti-icing technologies’

Reviewer #2

General comments: The paper proposes a design of a three-layer membrane structure, which features surface hydrophobicity, high-temperature radiative heat dissipation performance, and low-temperature solar absorption performance, with anti-icing as its application scenario, and demonstrates its application value. The design idea of the paper's work has certain innovativeness, but some key issues need further demonstration and explanation.

General response: We sincerely appreciate the reviewer's thoughtful evaluation and constructive feedback on our manuscript. We are grateful that the reviewer acknowledges the innovative aspects of our proposed three-layer membrane structure. In response to the reviewer's valuable comments, we have carefully revised the manuscript to provide additional experimental evidence and detailed explanations addressing the key issues raised. Our point-by-point responses to all specific comments are provided below.

Comment 1: Phase change materials are used for heat storage to delay the icing time, and the delay time is related to the thickness of the phase change material coating. The thickness of the phase change coating used in the paper is limited, so the icing delay time in actual application scenarios is very short, which can hardly achieve practical anti-icing effects during long winter nights. How does the author consider this?

Response 1: We appreciate the reviewer's insightful comment. It is true that the delay time of icing is closely related to the thickness of the phase change material (PCM) layer. In this work, the coating thickness (~ 2 mm) was limited by experimental conditions, which indeed results in a relatively short icing delay (~640 s, Figure 4i). Our purpose here was to demonstrate the feasibility and fundamental mechanism of integrating PCMs with superhydrophobicity for anti-icing at cold night. For practical applications, the icing delay time can be further prolonged by thicker coatings or by optimizing the encapsulation method to enhance heat storage capacity. Additionally, combining PCM coatings with other active anti-icing strategies (such as low-power joule heating or waste heat recovery) offers a promising pathway to achieve long-term performance under nighttime conditions. **These aspects will be the focus of our future work and relevant discussion has been added to the revised manuscript and marked in blue (see Discussion section on page 21).**

[Revised contents in the revised Manuscript]

Page 21: *Another direction lies in integrating photothermal phase change materials with complementary active anti-icing strategies, such as low-power Joule heating or waste-heat recovery, to mitigate icing risks in the absence of solar irradiation.*

Comment 2: The hydrophilic-hydrophobic property change of the hydrogel is used to realize the switching between light reflection and light transmission. In the hydrophobic state, how to prevent the loss of released water? After water release, the volume of the hydrogel polymer will shrink; how to prevent or handle the impact of polymer deformation?

Response 2: We appreciate the reviewer's insightful comments regarding the stability of the hydrogel during the phase transition. We agree that the hydrogel will shrink after water release. In our design, the hydrogel was encapsulated into a mold comprised of a PET film and a MNTS film bonded together using double-sided tape. The edges of the mold were sealed with UV-curing glue to prevent the loss of released water (See **Methods section**). Therefore, our hydrogel can sustain multiple heating-cooling cycles without significant water loss and volume shrinkage (see **page 12 and Supplementary Fig. 24**).

Fig. R1 (Supplementary Fig. 24). Weight change of PND₅E₂₀ hydrogels during heating process (40 °C) for 10 hours.

Comment 3: How are the three layers combined, and what is the bonding force? How is the overall structure of the three-layer structure encapsulated? In addition to the possible loss of water when the hydrogel is hydrophobic, the phase change material will also flow away after melting, which are related to the overall encapsulation of the structure.

Response 3: We appreciate the reviewer’s valuable comments. The hydrogel interlayer is sandwiched between the MNTS top layer and the PET film, with the edges sealed using double-sided adhesive tape and further reinforced by UV-curable glue to prevent leakage. The PTPCC bottom layer is bonded to the PET substrate after plasma surface treatment, and their edges are also sealed with UV-curable glue to enhance adhesion. The interfacial bonding between PTPCC and PET can be further improved by slightly reducing the curing agent ratio of PDMS. Peel tests (ASTM D903) were conducted to evaluate the interlayer adhesion within the TAPSS structure, showing strong bonding between the layers, with maximum peel strengths of 384 N/m for the top two layers and 78 N/m for the bottom two layers. Compared to reported peel strengths of PDMS-based composite coatings, our peel strength is higher. For example, ~19 N/m for skin-adhesive GCD-PDMS films (Science Translational Medicine, 2023, 15(693): eabq1634) and ~60 N/m for CNT/PDMS conductive elastomer electrodes (Nanoscale, 2025, 17(14): 8624-8633). Additional encapsulation of the PTPCC layer was not applied, as the leakage tests (**Supplementary Fig. 30**) confirmed negligible PCM leakage due to the capillary confinement within the porous matrix. Nevertheless, further encapsulation of the PTPCC layer could be adopted to provide additional protection if needed. In accordance with the reviewer’s suggestion, more detailed descriptions of the layer integration and bonding procedures have been added on pages 7, 23 of the revised manuscript and in Supplementary Fig. 7.

Fig. R2 (Supplementary Fig. 7). Peel force versus displacement for MNTS film-PNDE hydrogel and PET film-PTPCC interfaces.

[Revised contents in the revised Manuscript]

Page 7: *To ensure robust interfacial bonding among the multilayer components, several strategies were employed. The PET substrate, which serves both as the encapsulation layer for the hydrogels and the bonding interface with the PTPCC layer, was plasma-treated to promote adhesion. The curing agent ratio of PDMS was further optimized to strengthen its bonding with the PET substrate. Additionally, adhesive tape and UV-curable glue was applied along the edges to reinforce adhesion and prevent hydrogel leakage. Peel tests confirmed that the MNTS film and PNDE hydrogel were effectively bonded with a maximum peel strength of 384 N/m, while the adhesion between the PET film and PTPCC layer reached 78 N/m (Supplementary Fig. 7).*

Page 23: *Prior to bonding, the PET substrate was treated by plasma to improve surface energy, and the ratio of curing agent was optimized to enhance the adhesion of PTPCC to the PET substrate. After complete curing, the edges between PET and PTPCC were sealed with UV-curing glue to further strengthen the interfacial bonding. This bonding approach avoids covering the internal areas of the layers, thus having negligible influence on their optical and thermal performance. Subsequently, a 0.07 mm-thick MNTS film and the PET film were positioned on the top and bottom, respectively, and bonded together using double-sided tape to form a mold for encapsulating PNDE hydrogels. The edges of the mold were further sealed with UV-curing glue to prevent leakage and improve adhesion.*

Comment 4: The surface material has good radiative performance, which can exchange heat with outer space to reduce the material's temperature in summer. However, radiative cooling materials without temperature self-adaptive function still have cooling capacity at low temperatures in winter, especially during winter nights, radiative cooling will further lower the material's temperature, increasing the risk of icing. How to overcome this?

Response 4: We sincerely thank the reviewer for this valuable comment. We agree that the high emissivity of our MNTS film may result in excessive cooling during winter nights, thereby increasing the risk of icing. Indeed, it remains a great challenge to simultaneously tune both the solar and mid-infrared spectral responses. To address this limitation, we suggest replacing the MNTS film with an infrared-transparent polyethylene substrate coated with a low-emissivity Ag nanowire (AgNW) layer, as demonstrated in previous studies (Nature Communications, 2025, 16(1), 6952; Science Advances, 2022, 8(17), eabn7359). At elevated temperatures, the water released from the hydrogel can spread over the AgNW coating, effectively increasing the emissivity and enhancing radiative cooling. In

contrast, at lower temperatures, the released water is reabsorbed by the hydrogel, thereby reducing emissivity and suppressing excessive cooling. This temperature-adaptive modulation mechanism provides a promising pathway to mitigate the risk of icing while maintaining effective cooling performance in hot conditions. In addition, phase change materials that stores thermal energy during daytime and releases it during nighttime can help to delay icing during winter nights. According to your comment, we added relevant discussion in the revised manuscript on pages 21 (marked in blue).

Fig. R3. Intelligent window design with full-spectrum modulation capability (Nature Communications, 2025, 16(1), 6952).

[Revised contents in the revised Manuscript]

Page 21: *Beyond material substitution, dynamically tuning mid-infrared emissivity in response to ambient temperature remains a critical challenge for balancing radiative cooling and anti-icing performance during nighttime. This can be realized using an infrared-transparent polyethylene substrate coated with a low-emissivity Ag nanowire layer to encapsulate hydrogel, thereby suppressing excessive cooling under low-temperature conditions.*

Comment 5: The material preparation process is complex, and the price of raw materials is high. What is the prospect of large-scale preparation of the material?

Response 5: We sincerely thank the reviewer for raising this insightful question. We acknowledge that the current preparation process involves relatively complex steps and that certain raw materials (e.g., the transparent superhydrophobic top layer) are not yet optimized for large-scale production. In this study, our primary goal was to demonstrate the feasibility and effectiveness of the proposed

concept at the laboratory scale. For future practical applications, we see several promising directions to improve scalability and reduce cost:

1) **Process simplification:** Current synthesis steps of transparent superhydrophobic film and photothermal film can be replaced by scalable coating methods (e.g., spray-coating, roll-to-roll processing), which are already widely adopted in industrial production (Journal of Vacuum Science & Technology B, 2015, 33(6)).

2) **Material substitution:** The use of more abundant and cost-effective precursors that retain similar functional properties will significantly reduce the overall cost. For example, we can replace expensive multi-walled carbon nanotubes with carbon black and biomass-derived carbon materials.

Therefore, while this work focuses on proof-of-concept demonstration at the laboratory scale, we believe there is a clear pathway toward cost reduction and process scale-up, which will be the focus of our future research. Relevant discussion has been added to the revised manuscript and marked in blue (see Discussion section on page 21).

[Revised contents in the revised Manuscript]

Page 21: *The multilayer architecture could be simplified by dispersing selective photothermal nanomaterials (e.g., cesium tungsten bronze) within hydrogel matrices; scalability could be enhanced through spray-coating and roll-to-roll UV-NIL techniques; and costs could be reduced by employing more abundant and cost-effective precursors without compromising functionality.*

Reviewer #3:

General comments: This paper presents a development of a temperature-adaptive photothermal storage superhydrophobic film with a trilayer design, comprising a photothermal phase-change base layer, a freeze-resistant thermochromic hydrogel interlayer and a transparent superhydrophobic top layer. This study is expected to provide a useful wearable large area rapid thermal management and this can be recommended for publication if the authors can successfully respond to the following comments.

General responses: We sincerely appreciate the reviewer's positive and encouraging assessment of our work. We are pleased that the reviewer recognizes the potential of our temperature-adaptive photothermal storage superhydrophobic film with a trilayer design. Following the reviewer's valuable suggestions, we have carefully revised the manuscript and provide detailed responses to all specific comments. We believe that these revisions have further enhanced the rigor and readability of the paper. Point-by-point responses are provided below.

Comment 1: Diverging from conventional materials with constrained functionality, the suggested design combines a broadband high-absorptance photothermal phase-change base layer with a freeze-resistant thermochromic hydrogel interlayer that exhibits thermal-comfort transition temperature and strong visible-near-infrared (VIS-NIR) modulation capability. The integration between tri-layers will need integration. The integration layer for attaching with some chemicals will affect the optical and thermal performance between layers. This should be briefly discussed in the manuscript because it will affect the resultant performance.

Response 1: We appreciate the reviewer's insightful comment. In our design, the tri-layer structure was assembled using a minimal amount of UV-curable adhesive and double-sided tape only along the edges to ensure mechanical integrity. This bonding approach avoids covering the internal areas of the layers, thus having negligible influence on their optical and thermal performance. *We have added a brief clarification in the revised manuscript to explain this point, which was marked in blue (see methods section on page 23).*

[Revised contents in the revised Manuscript]

Page 23: *Prior to bonding, the PET substrate was treated by plasma to improve surface energy, and the ratio of curing agent was optimized to enhance the adhesion of PTPCC to the PET substrate.*

After complete curing, the edges between PET and PTPCC were sealed with UV-curing glue to further strengthen the interfacial bonding. This bonding approach avoids covering the internal areas of the layers, thus having negligible influence on their optical and thermal performance. Subsequently, a 0.07 mm-thick MNTS film and the PET film were positioned on the top and bottom, respectively, and bonded together using double-sided tape to form a mold for encapsulating PNDE hydrogels. The edges of the mold were further sealed with UV-curing glue to prevent leakage and improve adhesion.

Comment 2: The photothermal performance of PTPCC base layer determines the deicing efficiency of TAPSS film, which was optimized by tuning MWCNTs loading. At 2 wt% MWCNTs content, a broadband solar absorptance of nearly 97% was achieved (Fig. 4a) due to continuous energy levels of hybrid bonds. It is well understood the MWCNT will strongly absorb the sunlight. However, I wonder how the sunlight will be absorbed for the non-transparent mode. I wonder if the hydrogel absorbs the sunlight or hydrophobic layer absorb sunlight. Even though they may absorb sunlight, it is expected to be weak.

Response 2: We appreciate the reviewer's insightful question regarding the light absorption mechanism of the TAPSS film in the non-transparent mode. As shown in our optical characterization (**Fig. 1e**), the overall solar absorptance of the TAPSS film in the heated state is approximately **30%**. The absorption mainly originates from three components: the transparent superhydrophobic top layer, the PNIPAM-based hydrogel interlayer, and the photothermal phase-change base layer. Specifically, the top layer primarily absorbs ultraviolet radiation (**Fig. 1d**), while the hydrogel exhibits notable absorption in the near-infrared region (**Fig. 3d, e**). Together, these two layers account for about **21%** of the total solar absorptance. In addition, the hydrogel allows approximately **9%** of the incident sunlight to transmit through, which is then almost entirely absorbed by the underlying base layer. Therefore, in the non-transparent mode, the upper two layers contribute significantly to solar-to-thermal energy conversion, while the photothermal base layer absorbs the remaining solar energy and further enhances heat generation. As a result, even in the non-transparent state, a temperature increase exceeding 10°C can be achieved under strong solar irradiation and hot weather conditions (**Fig. 6b**).

Comment 3: Continuing the previous comment, the general transient temperature measurement starting from the non-transparent mode needs to be provided. I guess the temperature evolution in Figure 4(i) is for the case right after the time when the bottom light absorbing layer is exposed. I

wonder how long it takes for the transition from non-transparent to transparent mode transition as soon as the sunlight start to irradiate (not just a cyclic test shown in figure 6(b)).

Response 3: We appreciate the reviewer's insightful comment. We have to clarify that **Fig. 4i** only presents the photothermal response and energy storage behavior of the photothermal phase change composite (PTPCC) layer, without involving the upper two layers. Therefore, the transition between the non-transparent and transparent modes is not reflected in this figure. We have added an explanation in the revised manuscript to avoid potential misunderstanding (see page 15).

In the field test (i.e., **Fig. 6b**), we did not specifically measure the transition time under sunlight irradiation, as it strongly depends on the ambient temperature and solar intensity. In our experimental conditions, the ambient temperature was relatively high (>27 °C), close to the hydrogel's transition temperature (~ 28 °C), suggesting that the phase transition would occur rapidly under sunlight exposure. As it takes only ~ 10 s for transparent-to-opaque transition when heated at 40 °C (**Supplementary Fig. 23**), the transition occurs within several minutes or even tens of seconds once the sunlight exposure begins.

Comment 4: The adaptive functionality of TAPSS film necessitates a thermochromic interlayer that exhibits high transparency in the cold state, strong solar modulation capability and a suitable transition temperature. Among various explored thermochromic materials including vanadium dioxide (ACS Appl. Mater. Interfaces 13, 22495-22504 (2021)) and perovskite (Adv. Funct. Mater. 31, 2010426 (2021)), thermochromic hydrogels, particularly poly(N-isopropylacrylamide) (PNIPAM)-based systems (J. Mater. Chem. A 2, 13550-13555 (2014)), outperform due to superior solar modulation ability and thermal responsiveness. Recent related progresses in the optical transparency change by phase change material (Sci. Adv., 5, eaav4916, (2019)), dynamic crosslinking of liquid crystal elastomer (Nat. Mater., 23, 834 (2024)) need to be briefly discussed in the introduction part to provide the readers with the recent related progresses.

Response 4: We appreciate the reviewer's valuable suggestion. Following the recommendation, the optical transparency modulation based on supersaturated salt hydrate crystal (Sci. Adv., 5, eaav4916, 2019) and the dynamic crosslinking strategy of liquid crystal elastomers (Nat. Mater., 23, 834, 2024) have been cited and discussed to highlight the latest developments in thermochromic materials. These additions help place our work in the broader context of recent progress in thermo-responsive materials.

The corresponding revisions can be found in the revised manuscript on **page 6** (marked in blue).

[Revised contents in the revised Manuscript]

Page 6: Among various explored thermochromic materials including vanadium dioxide (VO_2)^{Error!}
Reference source not found., perovskite^{Error!} Reference source not found., supersaturated salt hydrate crystal^{Error!}
Reference source not found. and liquid crystal elastomer^{Error!} Reference source not found.,

Comment 5: Figure 1(e) presents the absorption spectra of the TAPSS film in heating and cooling modes. There are some issues to be cleared for this optical property measurement. The adaptive functionality of TAPSS film necessitates a thermochromic interlayer that exhibits high transparency in the cold state, strong solar modulation capability and a suitable transition temperature. I wonder if the optical property was measured with the ice formed on the surface because ice is basically water and water molecule will be strongly absorbed at some peaks in NIR wavelength.

Response 5: We appreciate the reviewer's valuable comment. The optical property measurements of the TAPSS film were conducted **without ice coverage** on the surface to ensure that the obtained spectra represent the intrinsic optical response of the film itself. We agree that ice or condensed water on the surface could introduce additional absorption features in the near-infrared region. It should be noted that since the thermochromic interlayer is based on a hydrogel containing water, certain infrared absorption peaks corresponding to the O-H vibrational modes of water molecules can still be observed in the near-infrared (NIR) wavelength (Nat. Commun., 2025, 16, 6952; Adv. Mater. 2023, 35(20), 2211716.). These features are intrinsic to the hydrogel component rather than caused by surface ice formation.

Comment 6: Continuing the previous comments, there seems to be still strong absorption over 1 micron wavelength even for the cooling mode. This may mean considerable amount of solar energy will be absorbed even for the cooling mode and it may induce an unexpected mode transition for the prolonged use under a certain condition. I wonder how this can be overcome.

Response 6: We appreciate the reviewer's insightful comment. We agree that the strong absorption of water in the NIR region is the main reason why the absorption of the TAPSS film cannot be further reduced in the cooling mode. This intrinsic absorption of H_2O limits the hot-state reflectivity of thermochromic hydrogels. Recent research (Nat. Commun., 2025, 16, 6952) has demonstrated that replacing H_2O with isotopic heavy water (D_2O) can significantly suppress the NIR light absorption,

thus enhancing the cold-state transmittance and hot-state reflectivity. In future work, such isotopic substitution could be explored to further improve the solar modulation and stability of the TAPSS film under prolonged solar exposure.

Fig. R1. Differences in the spectra of H₂O and D₂O under near-infrared (NIR) light (Nature Communications, 2025, 16(1), 6952).

Comment 7: Unlike conventional photothermal materials, our TAPSS film not only enables efficient anti-/deicing in winter but also prevents overheating in summer due to its exceptional solar regulation capability. This dual functionality can significantly reduce building energy consumption while maintaining year-round thermal comfort. I wonder where the cooling functionality comes from. Is it just reflection effect or is there any radiative cooling effect?

Response 7: We appreciate the reviewer's insightful comment. The cooling functionality of the TAPSS film originates from the combined effects of high solar reflectance and moderate radiative cooling. Specifically, the top transparent superhydrophobic layer exhibits high mid-infrared emissivity (see Fig. 2e), while the hydrogel interlayer possesses strong solar reflectivity in the hot conditions (see Fig. 3d). This synergistic structure enables effective suppression of solar heating and partial heat dissipation through thermal radiation. However, we would like to clarify that the TAPSS film can not achieve sub-ambient radiative cooling during daytime (see Fig. 6b).

Comment 8: The authors proposed a temperature-adaptive photothermal storage superhydrophobic film featuring a rationally engineered multilayer structure by separating solar absorption layer and light modulation layer etc. Similar approach has been applied to the tunable radiative cooling (Small, 20, 2308572 (2024); Nanophotonics, 13, 543 (2024)) and to maximize photothermal absorption on the multifunctional multilayered structures (Joule, 9, 102007 (2025); Small, 20, 2304338 (2024)) for radiative cooling. They need to be briefly discussed in the introduction part to provide the readers with the recent related progresses.

Response 8: We appreciate the reviewer's valuable suggestion. In the revised manuscript, we have added a brief discussion in the **Introduction** to include recent related studies on multilayer designs for tunable radiative cooling and solar heating (marked in blue, see page 4). By referencing these studies, we aim to better highlight the relevance and novelty of our design within the broader context of self-adaptive photothermal anti-/deicing.

[Revised contents in the revised Manuscript]

Page 4: Furthermore, several Janus-structure materials^{Error! Reference source not found.} with tunable radiative cooling and solar heating have been proposed for season-adaptive thermal regulation, whereas they must be manually flipped and lack passive anti-icing capabilities.

Comment 9: To preserve the photothermal performance of the underlying layer while delaying ice formation, the top layer must simultaneously achieve high transparency and superhydrophobicity, which was fabricated via ultraviolet nanoimprint lithography (UV-NIL, Supplementary Fig. 1) to precisely engineer subwavelength nanostructures. Nanoimprint lithography usually suffers from the large area production yet. I wonder what is the largest size can be produced now and what is the limiting factor and how can it be overcome?

Response 9: We appreciate the reviewer's valuable comment. Currently, the TAPSS film can be fabricated up to a size of approximately 10 cm×10 cm (see Fig. 1b), which is mainly limited by the dimension of the anodic aluminum oxide (AAO) template used in the UV-NIL process. The scalability of nanoimprint lithography indeed remains a challenge for large-area production. To address this limitation, roll-to-roll nanoimprint techniques (Journal of Vacuum Science & Technology B, 2015, 33(6)) are considered a promising approach to achieve continuous and large-scale fabrication. We have discussed this limitation and the possible solutions in the Discussion section of the revised

manuscript (marked in blue, see page 21).

[Revised contents in the revised Manuscript]

Page 21: *scalability could be enhanced through spray-coating and roll-to-roll UV-NIL techniques; and costs could be reduced by employing more abundant and cost-effective precursors without compromising functionality.*

Comment 10: I wonder if the same sample shows different performance depending on the thermal property of the surface where the sample is mounted, for example, on the wood or on the metal surface.

Response 10: We appreciate the reviewer's valuable question. We believe that the substrate's thermal properties have a relatively minor influence on the anti-icing performance, since the icing behavior is primarily governed by the surface wettability. Meanwhile, both the photothermal de-icing and radiative cooling performances are mainly determined by the spectral characteristics of the material, and the heat transfer effect from the substrate is limited. Therefore, our temperature-adaptive photothermal storage superhydrophobic film is expected to maintain stable performance across a wide range of substrate materials.

Comment 11: Some of the supplementary movies are not real time. The playing speed should be indicated on the whole movie clips (not just the title pages).

Response 11: We appreciate the reviewer's careful observation. The playback speed of each supplementary movie has now been clearly indicated throughout the entire video clips, rather than only on the title pages.

Comment 12: The general usage of language including typos and grammatical errors need to be checked again.

Response 12: We appreciate the reviewer's careful reminder. The entire manuscript has been thoroughly rechecked to correct minor grammatical issues, typographical errors, and inconsistencies in wording. The revised version has improved language accuracy and readability throughout. For example,

(1) **On page 3**, the 'among various surfaces' was corrected to 'among various passive strategies'.

- (2) **On page 4:** the ‘electric systems’ was corrected to ‘electrical systems’.
- (3) **On page 6:** the ‘dominant amide-water hydrogen bonding’ was corrected to ‘dominant hydrogen bonding between amide groups and water molecules’.
- (4) **On page 15:** ‘The’ was added before the ‘TAPSS film demonstrated exceptional freezing delay’.
- (5) **On page 20:** The ‘highlighting the practicality for scalable applications’ was corrected to ‘highlighting its practicality for scalable applications’
- (6) **On page 21:** the ‘one pot process’ was corrected to ‘one-pot process’.
- (7) **On page 22:** the ‘where x was the weight percentage of DMAA to NIPAM and y was the weight percentage of EG to water’ was corrected to ‘where x represents the weight percentage of DMAA to NIPAM and y represents the weight percentage of EG to water.’

These corrections were marked in blue in the revised manuscript.

Comment 13: Some of the digital pictures are missing scale bars. They should be added to the pictures.

Response 13: We appreciate the reviewer’s careful reminder. Scale bars have been added to the digital images in **Fig. 1c**, **Fig. 2a** and **Fig. 4d** to clearly indicate the spatial dimensions.

Reviewer #4:

General comments: This manuscript presents a temperature-adaptive photothermal storage superhydrophobic (TAPSS) film designed to mitigate the overheating issue commonly observed in traditional photothermal surfaces during hot seasons. The TAPSS film features an innovative trilayer structure, comprising a photothermal phase-change base layer, a freeze-resistant thermochromic hydrogel interlayer, and a transparent superhydrophobic top layer. This design enables the film to exhibit excellent anti-/deicing performance under cold conditions, along with remarkable radiative cooling performance in hot environments. Additionally, the film demonstrates strong UV resistance and mechanical durability, highlighting its potential for practical applications in diverse scenarios. Overall, the manuscript is innovative and supported by comprehensive experimental data, representing a notable advancement in the field of intelligent anti-/deicing technology. It is recommended for publication after addressing the following issues.

General responses: We sincerely thank the reviewer for the positive and encouraging evaluation of our work. We are pleased that the reviewer recognizes the innovation, comprehensive experimental validation, and practical potential of our TAPSS film. In response to the reviewer's valuable suggestions, we have carefully revised the manuscript to clarify descriptions, strengthen the supporting evidence, and improve the overall presentation. We believe that these revisions have further enhanced the quality and readability of the paper. Detailed, point-by-point responses to each comment are provided below.

Comment 1: According to Fig. 1e, the solar modulation ability decreases significantly at wavelengths beyond 1.2 μm . What is the underlying reason for this reduction? Could the authors propose potential solutions, such as employing alternative thermochromic hydrogels, to enhance the solar modulation performance in this spectral range?

Response 1: We appreciate the reviewer's insightful comment. The significant reduction in solar modulation ability beyond 1.2 μm primarily arises from the intrinsic absorption of water in the near-infrared (NIR) region, which results in minimal spectral difference between the cold and hot states of the hydrogel. Recent research (Nat. Commun., 2025, 16, 6952) has shown that substituting H_2O with isotopic heavy water (D_2O) can effectively suppress NIR absorption, thereby improving the cold-state transmittance and hot-state reflectivity. In future work, such isotopic substitution could be investigated to further enhance the solar modulation performance of the TAPSS film in the NIR range.

Comment 2: The authors used SEM images and pore size distribution data to explain that the opacity of thermochromic hydrogels above the LCST arises from enhanced light scattering. It would be beneficial to include additional quantitative evidence, such as light scattering efficiency simulations, to further substantiate this mechanism.

Response 2: We thank the reviewer for this valuable suggestion. Following the comment, we performed Monte Carlo simulations based on the pore size distribution derived from the SEM analysis to quantitatively evaluate the light scattering behavior of the hydrogel. The calculated reflectance (see **Fig. R1**) of the hydrogel before and after the phase transition exhibits a consistent trend with our experimental results (see **Fig. 3d**), confirming that the increased opacity above the LCST originates from enhanced light scattering. In the simulation, the porosity was assumed to be 0.2 and the refractive index was set to 1.33. The minor discrepancy between the experimental and simulated data may result from variations in the actual porosity and refractive index of the material.

Fig. R1. Simulated reflectivity spectra of thermochromic hydrogels before and after phase transition.

Comment 3: In Fig. 3a and 3b, the enthalpy peak associated with the phase change process appears to diminish with the addition of ethylene glycol. What explains this reduction in phase change enthalpy?

Response 3: We appreciate the reviewer's valuable comment. The observed reduction in phase change enthalpy with increasing ethylene glycol (EG) content can be attributed to the alteration of the hydrogen-bonding network between the polymer and the solvent. Specifically, the hydrogen

bonds formed between water and EG molecules are stronger and more stable than those between PNIPAM and either water or EG. As the EG concentration increases, a larger fraction of water molecules preferentially interacts with EG rather than the polymer chains, leading to a reduction in the number of PNIPAM–solvent hydrogen bonds. Consequently, the extent of the polymer–solvent cooperative rearrangement during the phase transition decreases, resulting in a lower phase change enthalpy.

Comment 4: In Fig. 4f, the cooling rates of CPCM-80 and PTPCC are quite similar, yet the heating rate of CPCM-80 is considerably slower than that of PTPCC. Could the authors elaborate on the mechanism responsible for this difference?

Response 4: We appreciate the reviewer’s insightful comment. The slower heating rate observed for CPCM-80 compared to PTPCC is primarily due to the surface whitening that occurs upon phase change material (PCM) solidification (**Supplementary Fig. 6**). This whitening significantly reduces the solar absorptance of CPCM-80, thereby suppressing its photothermal conversion efficiency and leading to a slower temperature rise. In contrast, the PTPCC incorporates a photothermal layer on top of the CPCM-80, which effectively prevents surface whitening and maintains strong solar absorption, resulting in a faster temperature increase.

Fig. R2 (Supplementary Fig. 6). Photographs of the **a** composite phase-change material (CPCM) and **b** PTPCC after placing in a cold chamber at $-20\text{ }^{\circ}\text{C}$.

Comment 5: The deicing time of the TAPSS film is somewhat longer compared to other photothermal surfaces reported in the literature, which may be attributed to the thermal resistance of the upper layers and the thermal energy storage behavior of the bottom layer. What strategies could be proposed to accelerate the ice-melting process?

Response 5: We appreciate the reviewer's constructive comment. The relatively longer deicing time of the TAPSS film can indeed be attributed to the thermal resistance introduced by the upper nanostructured layer and the thermal energy storage effect of the bottom PCM layer, which together slow the surface temperature rise. To accelerate the ice-melting process, optimizing the thickness and thermal conductivity of the upper layers can be adopted. Specifically, the thermal conductivity of hydrogel could be enhanced by introducing nanoparticles with high thermal conductivity and constructing highly ordered hierarchical structures (Materials & Design, 2023, 233, 112239; Advanced Functional Materials, 2025, e18948). According to your comment, we have added detailed strategies to accelerate the ice-melting process on **pages 17** of the revised manuscript.

[Revised contents in the revised Manuscript]

Page 17: *The thermal conductivity of hydrogel can be further improved by introducing nanoparticles with high thermal conductivity and constructing highly ordered hierarchical structures.*

Comment 6: How is the adhesion between the bottom and middle layers ensured to prevent delamination during operation? Are there specific interfacial treatments or material modifications employed to enhance layer integration?

Response 6: We appreciate the reviewer's valuable comment. To ensure strong interfacial adhesion between the bottom and middle layers and prevent delamination during operation, several strategies were implemented. The PET substrate, serving as the bonding interface with the PTPCC layer, was plasma-treated to enhance its surface energy and promote chemical bonding. The PDMS curing agent ratio was further optimized to improve interlayer adhesion. In addition, adhesive tape and UV-curable glue were applied along the film edges for mechanical reinforcement. Peel tests (ASTM D903) confirmed robust interfacial bonding, with the adhesion strength between the PET film and PTPCC layer reaching 78 N/m (**Supplementary Fig. 7**). Compared to reported peel strengths of PDMS-based composite coatings, our peel strength is higher. For example, ~19 N/m for skin-adhesive GCD-PDMS films (Science Translational Medicine, 2023, 15(693): eabq1634) and ~60 N/m for CNT/PDMS conductive elastomer electrodes (Nanoscale, 2025, 17(14): 8624-8633). Following the reviewer's suggestion, we have added more detailed descriptions of the interfacial treatment and layer integration procedures on **pages 7 and 23** of the revised manuscript, as well as in **Supplementary Fig. 7**.

Fig. R3 (Supplementary Fig. 7). Peel force versus displacement for MNTS film-PNDE hydrogel and PET film-PTPCC interfaces.

[Revised contents in the revised Manuscript]

Page 7: *To ensure robust interfacial bonding among the multilayer components, several strategies were employed. The PET substrate, which serves both as the encapsulation layer for the hydrogels and the bonding interface with the PTPCC layer, was plasma-treated to promote adhesion. The curing agent ratio of PDMS was further optimized to strengthen its bonding with the PET substrate. Additionally, adhesive tape and UV-curable glue was applied along the edges to reinforce adhesion and prevent hydrogel leakage. Peel tests confirmed that the MNTS film and PNDE hydrogel were effectively bonded with a maximum peel strength of 384 N/m, while the adhesion between the PET film and PTPCC layer reached 78 N/m (Supplementary Fig. 7).*

Page 23: *Prior to bonding, the PET substrate was treated by plasma to improve surface energy, and the ratio of curing agent was optimized to enhance the adhesion of PTPCC to the PET substrate. After complete curing, the edges between PET and PTPCC were sealed with UV-curing glue to further strengthen the interfacial bonding. This bonding approach avoids covering the internal areas of the layers, thus having negligible influence on their optical and thermal performance. Subsequently, a 0.07 mm-thick MNTS film and the PET film were positioned on the top and bottom, respectively, and bonded together using double-sided tape to form a mold for encapsulating PNDE hydrogels. The edges of the mold were further sealed with UV-curing glue to prevent leakage and improve adhesion.*

Manuscript Number: NCOMMS-25-63266A

Response Letter

Reviewer #1:

General comments: I thank the authors for diligently responding to all of my comments and concerns. In general I think the manuscript is greatly improved, and would be suitable for publication. Two small comments that the authors may wish to consider prior to final publication

General responses: We sincerely appreciate the reviewer's positive evaluation of our revised manuscript and are grateful for the acknowledgment of our efforts in addressing all previous comments and concerns. We are pleased to hear that the reviewer finds the manuscript greatly improved and suitable for publication. We have carefully considered the two additional suggestions and have incorporated the corresponding revisions in the updated manuscript. We thank the reviewer again for the constructive feedback and valuable time devoted to our work. Our point-by-point responses to these two comments are provided below.

Comment 1: The tetradecane immersion test is poorly described in your methods and not really a great demonstration of resistance to contamination, as I assume the oil was washed off prior to testing the subsequent superhydrophobicity? Since the surface was clearly wet by tetradecane it indicates the surface is not superoleophobic, which is ok as that was not the goal of your study, but again is a limitation that could be recognized rather than hid / downplayed. This is also another reason using PFAS is not required, as typically the only reason PFAS is necessary is to achieve things like superoleophobicity.

Response 1: We thank the reviewer for this constructive comment. We agree that the tetradecane immersion test required clearer explanation. In the revised Supplementary Information (page 2), we now provide a more detailed description of the procedure, including the rinsing step prior to evaluating the superhydrophobicity. As correctly pointed out, the surface is clearly wetted by tetradecane and therefore is not superoleophobic. We have revised the manuscript to explicitly

acknowledge this limitation and to clarify that the coating exhibits only moderate oleophobicity, which is insufficient for self-cleaning or strong resistance to oil contamination. In line with the reviewer's suggestion, we have also noted that superoleophobicity typically requires PFAS-based chemistries, which are not necessary for (nor targeted in) our design. *These revisions have been incorporated into the Results on page 8 and are highlighted in blue.*

[Revised contents in the revised Manuscript]

Pages 8: *In contrast, the MNTS film exhibited only moderate oleophobicity toward tetradecane (CA~95°), and therefore cannot provide self-cleaning against oil contamination. Nevertheless, after rinsing, the surface retained stable superhydrophobicity after being immersed in tetradecane for 120 h.*

[Revised contents in the revised Supplementary Information]

Page 2: *The CA and SA of rinsed surface were measured after certain time intervals.*

Comment 2: There is still a disconnect between all the applications / uses that the authors claim the coating will be good for, and the more practical discussion given in the Response file. For example, if the coating is just a demonstration and proof of concept, it will not potentially find any real world use or have any potential benefit to roofs, aircraft, etc. Right now the authors are trying to 'have their cake and eat it too', ie claim that the coating will indeed save the world, but also that's it's just a proof of concept and so all the shortcomings can be ignored.

Response 2: We thank the reviewer for this important and constructive comment. We agree that the previous description may have inadvertently created the impression that our coating is both an early-stage proof-of-concept and, at the same time, immediately suitable for broad real-world deployment. To avoid this confusion, we have substantially revised the manuscript to more clearly delineate the scope and practical relevance of our work.

In the revised text, we now explicitly state that the TAPSS film is presented as a proof-of-concept platform that integrates superhydrophobicity, thermochromism, and photothermal energy storage to enable self-regulated photothermal anti-/deicing. While the combined mechanism and material architecture may inform future adaptive photothermal materials for seasonal thermal management, we acknowledge that the current formulation is not yet engineered for real-world implementation on

roofs, aircraft, or other large-scale infrastructure. Critical aspects (e.g., long-term durability under outdoor exposure, adhesion under dynamic loading, and large-area scalability) would require further development before any practical deployment can be envisioned. To ensure that this distinction is clear, we have added a concise explanatory sentence in the Discussion section (page 16), explicitly outlining these limitations and clarifying the intended contribution of the present work. We hope that these revisions address the reviewer's concern and provide a more balanced and accurate framing of our study.

[Revised contents in the revised Manuscript]

Page 16: *We anticipate that this proof-of-concept will inspire the design of adaptive photothermal materials, while substantial opportunities remain to further enhance mechanical robustness and long-term durability under complex operating conditions encountered in practical applications.*